# GROUP-NORMALIZED IMPLICIT VALUE OPTIMIZATION FOR LANGUAGE MODELS

**Yunseon Choi**    **Junyoung Jang**    **Chaeyoung Oh**    **Minchan Jeong**
**Doohwan Hwang**    **Kee-Eung Kim**
KAIST AI
{cys9506, kekim}@kaist.ac.kr

## ABSTRACT

Fine-tuning Large Language Models (LLMs) with reinforcement learning (RL) has become a key technique for enhancing performance on a wide range of tasks, from user alignment to complex reasoning. However, this approach is often hindered by the difficulty of fine-grained credit assignment, as it typically relies on sparse rewards given only at the end of a completely generated sequence. Conventional solutions often require training an auxiliary value network known as critic, which introduces significant computational overhead and training instability. We present *Group-Normalized Implicit Value Optimization (GN-IVO)*, a novel, critic-free algorithm that directly addresses this challenge. GN-IVO learns step-level values implicitly from the policy through a group-normalized distributional matching objective. This approach elegantly circumvents the need for an explicit critic and avoids the computation of the intractable partition function by normalizing values across a group of sampled model responses. Theoretically, we prove that our objective recovers the true value function up to a constant, guaranteeing that the optimal policy is preserved. We demonstrate the practical effectiveness of GN-IVO on a diverse set of text generation and reasoning tasks, showing that it consistently outperforms strong RL baselines for LLMs.

## 1 INTRODUCTION

Reinforcement Learning (RL) has emerged as a powerful paradigm for post-training Large Language Models (LLMs) (Ouyang et al., 2022; Kumar et al., 2025). Its success is well-established across a range of critical applications, from aligning model outputs with human preferences (Bai et al., 2022; Rafailov et al., 2023) to enhancing their complex, multi-step reasoning abilities (Zelikman et al., 2024; DeepSeek-AI, 2025). Yet, many RL approaches for LLMs (Rafailov et al., 2023; Ethayarajh et al., 2024; Ahmadian et al., 2024) cast policy learning as a contextual bandit, where the entire generated response is treated as a single action that is assigned a scalar score by a reward model. This formulation clashes with the inherently sequential, long-horizon nature of both token-by-token generation and step-by-step reasoning, in which intermediate decisions compound over time. Moreover, outcome-only rewards yield a sparse and delayed learning signal, offering little guidance about which token or step-level decisions drive the final quality. As a result, step-level credit assignment remains opaque, suggesting that language generation should be treated as sequential decision making rather than a one-shot bandit.

Recent works (Zeng et al., 2024; Wang et al., 2024; 2025) therefore model LLM generation as a Markov Decision Process (MDP) to incorporate the sequential decision making. A common approach is to enforce Bellman consistency and learn step-aware value estimates. In particular, Wang et al. (2025); Liu et al. (2024) train policies within the soft Q-learning framework (Haarnoja et al., 2017), parameterizing the policy as a Boltzmann distribution over Q-values with a partition function. However, because the partition function must still be estimated, these methods retain an auxiliary estimator. While this estimator can serve as a baseline for variance reduction in policy updates, this introduces practical overhead; additional networks to train and tune, increased computational and memory costs, and added complexity in the post-training pipeline. This trade-off, gaining step-aware feedback at the expense of a more complex training procedure, motivates methods that enable effective sequential learning without auxiliary networks or substantial engineering burden.

In this work, we propose *Group-Normalized Implicit Value Optimization (GN-IVO)*, an RL fine-tuning method that incorporates the sequential nature of language generation without requiring any auxiliary network. We first generalize the canonical KL-regularized objective for LLMs to partial sequence, deriving an explicit link between the policy and a soft value function. This value function quantifies the contribution of a partial sequence to the eventual outcome reward. To learn these values, we introduce a *group-normalized* objective. For a given query, we sample a group of complete responses and form a target distribution over them, proportional to their exponential rewards. We then match this target with the normalized model's predicted values for the partial sequence. This distributional matching forces the model to align its value predictions with the ground-truth relative values. We prove that this objective recovers the true value function up to an additive constant, which preserves the action preferences needed for the optimal policy. Finally, inspired by recent advances like DPO (Rafailov et al., 2023) that bypass explicit reward models, we likewise bypass explicit value function modeling by the direct policy-value relationship.

The result is a lightweight and direct optimization scheme that avoids computing the intractable partition function: normalization within each sampled group cancels any additive constant including the partition function. Thus, our method provides fine-grained feedback on partial sequences without incurring the overhead of training a separate value estimator. We validate the proposed approach on a diverse set of text generation and reasoning tasks, including the Helpful assistant, TL;DR summarization, prompt generation for text-to-image models tasks, and challenging reasoning benchmarks. Across these scenarios, GN-IVO demonstrates its effectiveness in performance compared to standard RL fine-tuning methods for LLMs.

We summarize our main contributions as follows:

1. **New Fine-Tuning Objective:** We introduce Group-Normalized Implicit Value Optimization, a novel RL objective that incorporates step-level feedback into LLM post-training. Our method dispenses with any auxiliary value/critic model, instead using group-based normalization to infer the relative value of partial sequences.

2. **Theoretical Guarantee**: We present a theoretical analysis showing that our normalized objective learns the true value function up to a constant offset, which does not affect the optimal policy. This result guarantees that optimizing our objective yields the same optimal policy as the original RL problem, thereby ensuring policy optimality and consistency.

3. **Empirical Validation:** We demonstrate the effectiveness of our approach on multiple benchmarks, including text generation and complex reasoning tasks. Our experiments show that the proposed method achieves higher rewards than standard RL algorithms, highlighting its practical benefit in aligning LLMs with sparse feedback.

## 2   RELATED WORKS

**Reinforcement Learning from Human Feedback (RLHF)**    LLMs are often aligned with human preferences using RLHF, wherein a reward model trained on human feedback guides the policy optimization. PPO (Schulman et al., 2017) has become the canonical algorithm in early large-scale alignment works (Ouyang et al., 2022). Despite its success, PPO-based RLHF can be resource-intensive and operationally complex. It typically requires keeping multiple models in memory, a policy, a frozen reference model for KL control, a critic model, and a reward model. The performance is also sensitive to hyperparameters due to high variance of the policy gradient estimates and delayed outcome rewards. These challenges have motivated research into simpler fine-tuning methods that bypass the full RL machinery while aiming to preserve alignment quality. DPO (Rafailov et al., 2023) reframes the KL-regularized objective as a supervised loss on preference-labeled data, eliminating explicit reward model training, and thereby improving stability and computational efficiency. Numerous extensions build on this paradigm (Maeng et al., 2025; Qi et al., 2024). Distinct from these preference-based methods, DRO (Richemond et al., 2024) addresses scenarios where feedback is provided as a score for each completion, rather than as a pairwise preference. To handle this format, DRO incorporates the Soft Bellman equation into a bandit-setting objective, effectively learning from direct reward labels. BRAIN (Pandey et al., 2024) formulates RLHF as a distribution-matching problem between the target and learned policies, proposing a self-normalized baseline to reduce variance. However, DPO, DRO, BRAIN and most of their variants formulate the generation task as a bandit problem where generating a completion is considered a single action that receives a

single reward. This formulation obscures the sequential structure of language generation, thereby limiting performance on long-horizon tasks such as mathematical reasoning (Yuan et al., 2025).

**Reinforcement Learning for LLM Reasoning**  Beyond human preference alignment, recent works have explored RL to improve LLMs' reasoning abilities, which require generating long chains of intermediate steps. Such training imposes a substantial memory demand due to long generated trajectories, even when the base model is modest in size. Since the critic network adds to this memory cost, a key research direction has been to develop approaches that remove explicit value function learning in policy-gradient methods. For instance, GRPO (Shao et al., 2024) is a variant of PPO that dispenses with a learned critic and instead computes advantages from group-wise return statistics within each batch. By using the average return from a group of trajectories as a baseline, GRPO significantly reduces memory overhead while maintaining training stability. In a similar vein, RLOO (Ahmadian et al., 2024) extends REINFORCE with a leave-one-out baseline. It computes the advantage of each response by comparing its reward to the average reward of others in the group, thereby reducing variance without requiring a separate value predictor. Despite their improved memory-efficiency, methods like GRPO and RLOO still treat the entire generated sequence as a single action. By doing so, they fail to model the internal structure of multi-step reasoning and assign credit equally across all tokens based on a single, final reward. This simplistic treatment overlooks the chain-of-thought or stepwise dependencies within complex reasoning processes, making it difficult to pinpoint which parts of the generation led to a successful outcome. Recently, OREO (Wang et al., 2025) and DQO (Liu et al., 2024) have formulated reasoning steps as an MDP, employing soft Q-learning to learn the policy. However, the Bellman error minimization requires an additional value network, making the training procedure complex and memory-inefficient, which hinders scalability.

To bridge this gap, we introduce Group-Normalized Implicit Value Optimization (GN-IVO), a method designed to be both sequential and critic-free. Similar to GRPO and RLOO, GN-IVO utilizes a group of $K$ samples to remain critic-free. However, a key difference lies in how these samples are used: whereas GRPO computes advantages using batch statistics, GN-IVO uses the samples to form a group distributional matching objective. At the same time, and in spirit of OREO and DQO, our method explicitly models the step-by-step nature of generation to enable fine-grained credit assignment.

## 3 METHOD

### 3.1 BACKGROUND: KL-REGULARIZED POLICY OPTIMIZATION

KL-regularized policy optimization is a widely used approach for refining language model policies. The goal is to train a policy $\pi_\theta$ that maximizes the expected reward while staying close to a reference policy $\pi_{\theta_{\text{old}}}$, which is often the supervised fine-tuned (SFT) model that we start with. This objective is formalized as:

$$\max_\theta \mathbb{E}_{x \sim \mathcal{D}, y \sim \pi_\theta(\cdot|x)} \left[ R(x,y) - \alpha \log \frac{\pi_\theta(y|x)}{\pi_{\theta_{\text{old}}}(y|x)} \right] \tag{1}$$

where $x$ is a query sampled from the dataset $\mathcal{D}$, and $y$ is a completion generated by the policy $\pi_\theta$. The function $R(x,y)$ assigns a scalar reward to the completion $y$ for a given query $x$, and $\alpha \geq 0$ is a temperature coefficient. The optimal policy $\pi_{\theta*}$ for Eq. 1 has a known closed-form solution:

$$\pi_{\theta*}(y|x) = \frac{\pi_{\theta_{\text{old}}}(y|x) e^{R(x,y)/\alpha}}{Z(x)} \tag{2}$$

where $Z(x) = \mathbb{E}_{\pi_{\theta_{\text{old}}}(y|x)}[e^{R(x,y)/\alpha}]$ serves as the partition function to ensure the distribution is normalized. While the formulation in Eq. 1 is standard for RL post-training of LLMs, it relies on a single reward $R(x,y)$ assigned to the entire generated sequence $y$.

However, this single reward for the entire sequence provides a weak credit assignment signal. Given the autoregressive, token-by-token generation process of language models, a more granular credit-assignment scheme is warranted. We therefore reformulate the objective to incorporate expected rewards at each timestep, enabling denser supervision.

## 3.2 LEARNING STEP-LEVEL VALUES VIA GROUP NORMALIZATION

Our first key theoretical result establishes that the relationship in Eq. 2 extends to arbitrary sequence prefixes. Let $y = (y_0, \cdots, y_{T-1})$ be any sequence of length $T$, where $y_t$ denotes the token or reasoning step at timestep $t$. [1]

**Theorem 3.1.** *Suppose that $\pi_{\theta^*}$ and $\pi_{\theta_{old}}$ are autoregressive policies that generate sequences token-by-token, and that satisfy Eq. 2 for complete response $y$. For any $t \in \{1, \cdots T\}$, let $y_{<t} := y_{0:t}$ denote the length-$t$ prefix. Then, $y_{<t}$ distributions of these policies satisfy:*

$$\pi_{\theta^*}(y_{<t}|x) = \frac{\pi_{\theta_{old}}(y_{<t}|x)e^{V(x,y_{<t})}}{Z(x)} \tag{3}$$

*where a soft value function $V(x, y_{<t})$, which represents the expected future reward from $y_{<t}$, is defined as:*

$$V(x, y_{<t}) := \begin{cases} R(x,y)/\alpha & t = T, \\ \log \mathbb{E}_{\pi_{\theta_{old}}(y|y_{<t},x)}[e^{R(x,y)/\alpha}] & t < T. \end{cases} \tag{4}$$

While distributions over partial sequences are typically hard to analyze, Theorem 3.1 reveals an exact analytical form for $\pi_{\theta^*}(y_{<t}|x)$ with $V(x, y_{<t})$. The value function $V(x, y_{<t})$ provides a formal solution to the temporal credit assignment problem by quantifying the expected achievable reward given the tokens generated so far, $y_{<t}$. It allows the model to attribute the final monolithic reward back to each intermediate decision, enabling granular policy optimization.

**The challenge of evaluating the partition function** While the soft value function from Theorem 3.1 can be modeled with an auxiliary network $V_\psi$, doing so adds complexity and computational cost. Inspired by DPO (Rafailov et al., 2023) that bypasses an explicit reward model, we instead pursue a direct policy objective that avoids explicit value modeling. By rearranging Eq. 3, we can express the value function implicitly in terms of a policy ratio, which allows us to reframe value fitting as a policy optimization problem:

$$V(x, y_{<t}) = \log Z(x) + \log \frac{\pi_{\theta^*}(y_{<t}|x)}{\pi_{\theta_{old}}(y_{<t}|x)}. \tag{5}$$

A natural starting point for policy training loss is to minimize the mean-squared error (MSE) between the policy-defined value (Eq. 5) and the ground-truth value function (Eq. 4). Substituting our trainable policy $\pi_\theta$ for the unknown optimal policy $\pi_{\theta^*}$ yields the following loss function:

$$\mathcal{L}_{MSE}(Z, \theta) = \left(\log Z(x) + \log \frac{\pi_\theta(y_{<t}|x)}{\pi_{\theta_{old}}(y_{<t}|x)} - \log \mathbb{E}_{y \sim \pi_{\theta_{old}}(\cdot|y_{<t},x)}[e^{R(x,y)/\alpha}]\right)^2. \tag{6}$$

A practical difficulty in Eq. 6, however, is the presence of the partition function $Z(x)$. Its computation requires marginalizing over all possible sequences under $\pi_{\theta_{old}}$, which is intractable. While this term can be approximated with an auxiliary network, $Z_\phi(x)$, doing so introduces additional complexity and potential instability. This motivates a different approach that circumvents the need to estimate the partition function entirely.

**The group-normalized objective** To avoid an auxiliary network for the partition function, we propose a *group-normalized* objective for value estimation. The key insight is that for a fixed query $x$, the partition function $Z(x)$ is a constant across all candidate responses. By normalizing values over a sampled group of candidates, any shared constant including $\log Z(x)$ cancels out, making its explicit estimation unnecessary. We first develop this objective for an explicit estimator, $V_\psi(\cdot)$, before extending it to implicit, policy-only training.

The procedure starts by sampling a group of $K > 1$ i.i.d. completions, $\{y^{(i)}\}_{i=0}^{K-1}$, from $\pi_{\theta_{old}}(\cdot|x)$. For a set of prefixes $\{y_{<t}^{(i)}\}_{i=0}^{K-1}$ at a randomly sampled timestep $t \leq T$ [2], we propose the training

---

[1] We consider two types of $y_t$. For general text generation, $y_t$ is a single token, $y_t = w_t \in \mathcal{V}$. For reasoning tasks, $y_t$ denotes a complete reasoning step represented as a token subsequence $y_t = (w_{t_0}, \cdots, w_{t_{l_t-1}})$ with $l_t \geq 1$.

[2] We assume all $K$ completions have the same length $T$ after right-padding with [PAD] tokens. This is a mild assumption, as sequences are typically padded to a fixed maximum length in practice. Both the policy and reward models are designed to ignore padded positions, ensuring rewards are unaffected.

objective given by:

$$\min_{V_\psi} \mathbb{E}_{\substack{x\sim\mathcal{D},t\sim\mathrm{U}\{1,\dots T\},\\ y^{(0:K)},y_{<t}^{(0:K)}\sim\pi_{\theta_{\mathrm{old}}}(\cdot|x)}} \left[ -\sum_{i=0}^{K-1} \mathrm{e}^{R(x,y^{(i)})/\alpha} \log \frac{\mathrm{e}^{V_\psi(x,y_{<t}^{(i)})}}{\sum_{j=0}^{K-1} \mathrm{e}^{V_\psi(x,y_{<t}^{(j)})}} \right]. \tag{7}$$

Intuitively, this objective trains $V_\psi$ to assign higher values to those $y_{<t}$ that are likely to extend into high-reward outcomes. Considering the definition of $V(x, y_{<t})$ and the tower property, our objective is equivalent to minimizing a weighted group cross-entropy:

$$\min_{V_\psi} \mathbb{E}_{\substack{x\sim\mathcal{D},t\sim\mathrm{U}\{1,\dots T\},\\ y_{<t}^{(0:K)}\sim\pi_{\theta_{\mathrm{old}}}(\cdot|x)}} \left[ \Big(\sum_{j=0}^{K-1} \mathrm{e}^{V(x,y_{<t}^{(j)})}\Big) \cdot -\sum_{i=0}^{K-1} \frac{\mathrm{e}^{V(x,y_{<t}^{(i)})}}{\sum_{j=0}^{K-1} \mathrm{e}^{V(x,y_{<t}^{(j)})}} \log \frac{\mathrm{e}^{V_\psi(x,y_{<t}^{(i)})}}{\sum_{j=0}^{K-1} \mathrm{e}^{V_\psi(x,y_{<t}^{(j)})}} \right].$$

This equivalence reveals that the objective Eq. 7 frames the learning problem as one of distributional matching between two distributions defined over the group: (i) a target distribution derived from the normalized, exponential true values $V$, and (ii) a model distribution derived from the normalized, exponential estimated values of our learnable function $V_\psi$. From this observation, the following theorem establishes that the optimal solution to Eq. 7 recovers the true value function up to a constant offset.

**Theorem 3.2** (Consistency up to constant shift). *Assume unlimited model capacity and data. For any $K > 1$ and $t \in \{1, \cdots, T\}$, the minimizer $V_{\psi^*}$ of Eq. 7 recovers the soft value function $V$ of Eq. 4 up to an additive, $y_{<t}$-independent offset $C_t(x)$:*

$$V_{\psi^*}(x, y_{<t}) = V(x, y_{<t}) + \log C_t(x),$$

*equivalently* $\mathrm{e}^{V_{\psi^*}(x,y_{<t})} = C_t(x)\mathrm{e}^{V(x,y_{<t})}$.

This additive offset does not affect the resulting optimal policy, as in the following corollary.

**Corollary 3.3** (Policy invariance to additive shifts). *For any positive scalar $C_t(x)$, let $V'(x, y_{<t}) = V(x, y_{<t}) + \log C_t(x)$. The optimal policy for $V'$ remains the same as the optimal policy for $V$.*

Taken together, these results guarantee that the optimal policy derived from our learned value function $V_{\psi^*}$ is identical to the one obtained from the true value function $V$. The proofs are provided in Appendix A.

### 3.3 GROUP-NORMALIZED IMPLICIT VALUE OPTIMIZATION

The key step toward our final, a critic-free objective, is to eliminate the explicit value function $V_\psi$ entirely. Instead of training a separate network, we leverage the policy-value link from Theorem 3.2 by substituting the value term in Eq. 7 with its implicit, policy-defined equivalent.

From Theorem 3.2, we know that the optimal value function $V_{\psi^*}$ is equivalent to the true value $V$ up to a constant ($\mathrm{e}^{V_{\psi^*}} = C_t\mathrm{e}^V$). We can combine this with the policy-value link from Eq. 3, which states $\pi_{\theta^*}(y_{<t}|x) \propto \pi_{\theta_{\mathrm{old}}}(y_{<t}|x)\mathrm{e}^{V(x,y_{<t})}$. Together, these relationships allow us to express the exponential value function directly in terms of the policy:

$$\mathrm{e}^{V_{\psi^*}(x,y_{<t})} = C_t(x)Z(x)\frac{\pi_{\theta^*}(y_{<t}|x)}{\pi_{\theta_{\mathrm{old}}}(y_{<t}|x)}. \tag{8}$$

We now substitute this policy-defined expression for the exponential value function back into the model's predicted distribution within our objective (Eq. 7). The model's distribution is a softmax over the group of $K$ sampled values. By replacing each $\exp(V_\psi)$ term with its policy-defined equivalent from Eq. 8, the shared multiplicative terms $C_t(x)$ and $Z(x)$ cancel out from the numerator and denominator of the softmax operation:

$$\frac{\exp(V_\psi(y_{<t}^{(i)}))}{\sum_j \exp(V_\psi(y_{<t}^{(j)}))} \longrightarrow \frac{\cancel{C_t(x)Z(x)}\frac{\pi_\theta(y_{<t}^{(i)}|x)}{\pi_{\theta_{\mathrm{old}}}(y_{<t}^{(i)}|x)}}{\sum_{j=0}^{K-1}\cancel{C_t(x)Z(x)}\frac{\pi_\theta(y_{<t}^{(j)}|x)}{\pi_{\theta_{\mathrm{old}}}(y_{<t}^{(j)}|x)}} = \frac{\frac{\pi_\theta(y_{<t}^{(i)}|x)}{\pi_{\theta_{\mathrm{old}}}(y_{<t}^{(i)}|x)}}{\sum_{j=0}^{K-1}\frac{\pi_\theta(y_{<t}^{(j)}|x)}{\pi_{\theta_{\mathrm{old}}}(y_{<t}^{(j)}|x)}}.$$

This leads to our final, critic-free objective:

$$\mathcal{L}_{\text{GN-IVO}}(\theta) = \mathbb{E}_{\substack{x \sim \mathcal{D}, t \sim \text{U}\{1,...T\}, \\ y^{(0:K)} \sim \pi_{\theta_{\text{old}}}(\cdot|x)}} \left[ -\sum_{i=0}^{K-1} e^{R(x,y^{(i)})/\alpha} \left( \log \frac{\pi_\theta(y_{<t}^{(i)}|x)}{\pi_{\theta_{\text{old}}}(y_{<t}^{(i)}|x)} - \log \sum_{j=0}^{K-1} \frac{\pi_\theta(y_{<t}^{(j)}|x)}{\pi_{\theta_{\text{old}}}(y_{<t}^{(j)}|x)} \right) \right].$$
(9)

In our practical implementation, we replace $e^{R(x,y^{(i)})/\alpha}$ with the group-normalized weight $\frac{e^{R(x,y^{(i)})/\alpha}}{\sum_{j=0}^{K-1} e^{R(x,y^{(j)})/\alpha}}$ for stability.

**Training procedure** The training procedure for our algorithm follows the iterative scheme detailed in Algorithm 1. Each training iteration consists of data generation, reward evaluation, and policy optimization. First, we sample a batch of queries $x$ from the dataset $\mathcal{D}$. For each query, we generate a set of $K$ distinct responses by sampling from $\pi_{\theta_{\text{old}}}$, which is a frozen copy of the policy from the previous iteration. Next, each of these $K$ full-sequence responses is evaluated by the reward function $R(x,y)$ to obtain a scalar reward. Finally, these queries, responses, and rewards are used to perform a gradient update on the policy parameter $\theta$ by minimizing Eq. 9. After the policy $\pi_\theta$ has been updated, $\pi_{\theta_{\text{old}}}$ is updated to match $\pi_\theta$ for data generation in the next iteration.

---

**Algorithm 1** Group-Normalized Implicit Value Optimization

**Input:** Reward function $R$, learning rate $\eta$, the policy $\pi_\theta$ and set $\pi_{\theta_{\text{old}}} \leftarrow \pi_\theta$
**for** iterations **do**
    Sample a query $x \sim \mathcal{D}$ and generate $K$ responses $y^{(0:K)} \sim \pi_{\theta_{\text{old}}}(\cdot|x)$
    Evaluate reward $R(x, y^{(i)})$ for all $i \in \{0, \dots, K-1\}$
    Update $\theta$ by optimizing the following loss in Eq. 9, $\theta \leftarrow \theta - \eta \nabla_\theta \mathcal{L}_{\text{GN-IVO}}(\theta)$
    $\pi_{\theta_{\text{old}}} \leftarrow \pi_\theta$
**end for**

---

## 4 EXPERIMENTS

We evaluate our method on a diverse range of reasoning and text generation tasks. These evaluations include mathematical problem-solving with reasoning steps as well as open-ended language generation. The text generation tasks encompass creating helpful assistant responses, summarizing Reddit posts, and generating prompts for text-to-image models. Furthermore, we conduct a sensitivity analysis on hyperparameters.

### 4.1 MATHEMATICAL REASONING

This task assesses a model's ability to solve math problems that require multi-step reasoning, including generating intermediate derivations and the final answer.

**Experiment setup** For RL training on mathematical reasoning, we use the MATH dataset (Hendrycks et al., 2021) and employ Qwen2.5-Math-7B (Yang et al., 2024a) and Llama-3.1-8B-Instruct (Meta-AI., 2024) as backbone models. Input prompts include an instruction and few-shot examples to elicit step-by-step derivations, where each step is explicitly marked with a label (e.g. step1, step2). In this setting, we define $y_t$ as a single reasoning step; thus, $y_t$ is a sequence of tokens that forms a complete step, and $T$ is the total number of reasoning steps in the solution. The reward function is composed of two components, one for the correctness of the final answer and the other for adherence to the specified format. We evaluate performance on widely used mathematical reasoning benchmarks, including AMC 2023 (of Problem Solving, b), Minerva Math (Lewkowycz et al., 2022), Olympiad-Bench (He et al., 2024), and AIME 2024/2025 (of Problem Solving, a). These benchmarks cover a wide range of difficulties and reasoning depths, from competition-style problems to graduate-level questions. Further details are provided in the Appendix C.

**Metrics and decoding** We report performance using the Pass@1 and Pass@3 metric. For Pass@1, we generate responses via greedy decoding. For Pass@3, we sample three completions for each problem using a temperature of $0.7$. For all evaluations, we utilize the `math-verify` library to perform answer extraction and equivalence checking.

Table 1: Comparison of our method against baselines on the math reasoning task. The *Pass@3 (P@3)* metric is calculated over three trials per query. **Bold** and underline indicate the best and second-best results, respectively.

| Method | AMC2023 | | Minerva Math | | Olympiad-Bench | | AIME2024 | | AIME2025 | |
|---|---|---|---|---|---|---|---|---|---|---|
| | P@1 | P@3 | P@1 | P@3 | P@1 | P@3 | P@1 | P@3 | P@1 | P@3 |
| *Llama-3.1-8B-Instruct* | 27.5 | 37.5 | 25.7 | 32.7 | 15.6 | 24.2 | 3.3 | 10.0 | 0.0 | 0.0 |
| SFT-winning | 27.5 | 35.0 | 24.2 | 35.6 | 16.0 | 27.1 | 6.6 | 6.6 | 0.0 | 0.0 |
| Online DPO | 22.5 | 33.1 | 25.3 | 30.5 | 15.1 | 26.2 | 3.3 | 13.3 | 0.0 | 0.0 |
| PPO | 25.0 | 35.0 | 21.7 | 34.9 | 15.7 | 26.2 | 3.3 | **16.6** | **3.3** | 0.0 |
| DRO | 22.5 | 35.0 | 23.1 | 33.8 | 15.5 | 25.6 | 3.3 | 0.0 | 0.0 | 6.6 |
| OREO | 27.5 | 32.5 | 25.7 | 35.3 | 15.7 | 26.8 | 3.3 | 6.6 | 0.0 | 6.6 |
| RLOO | 35.0 | 40.0 | **26.1** | 34.1 | 17.9 | 26.1 | 6.6 | **16.6** | 0.0 | 0.0 |
| GRPO | 35.0 | 37.5 | 25.3 | 35.3 | **18.8** | 25.3 | 6.6 | **16.6** | **3.3** | 0.0 |
| Ours | **42.5** | **45.0** | **26.1** | **36.0** | 17.3 | **27.8** | **10.0** | **16.6** | **3.3** | 3.3 |
| *Qwen2.5-Math-7B* | 52.5 | 70.0 | 27.0 | 36.0 | 37.7 | 46.2 | 23.0 | 26.6 | 6.6 | 13.3 |
| SFT-winning | 57.5 | 62.5 | 30.5 | 36.0 | 38.8 | 49.0 | 23.3 | 26.6 | **13.3** | 13.3 |
| Online DPO | 57.5 | 70.0 | 27.2 | 37.1 | 36.2 | 49.2 | 23.3 | 30.0 | 10.0 | 13.3 |
| PPO | 47.5 | 67.5 | 28.3 | 37.8 | 38.6 | 49.0 | 23.3 | 30.0 | 10.0 | 13.3 |
| DRO | 55.0 | 67.5 | 31.2 | 37.1 | 37.7 | 48.7 | 23.3 | 33.3 | 10.0 | 13.3 |
| OREO | 55.0 | 70.0 | **31.6** | 38.6 | 38.5 | 49.1 | 16.6 | 30.0 | 10.0 | 13.3 |
| RLOO | 57.5 | 72.5 | 30.1 | 40.4 | 38.8 | 48.8 | 23.3 | 36.6 | **13.3** | 16.6 |
| GRPO | 60.0 | 70.0 | 29.7 | 37.8 | 39.2 | **49.4** | 26.6 | 33.3 | 6.6 | 13.3 |
| Ours | **62.5** | **75.0** | **31.6** | **41.9** | **39.8** | 49.0 | **30.0** | **40.0** | **13.3** | **23.3** |

**Baselines**    We compare our proposed method against the following baseline algorithms:

- **SFT-winning** generates two responses for each query and performs supervised fine-tuning on the response with the higher reward.

- **Online DPO** (Qi et al., 2024) extends DPO to an online setting. In each iteration, it generates two responses for each query, designates them as the winner and the loser based on their rewards, and uses this pair to compute the DPO loss in the policy update.

- **PPO** (Schulman et al., 2017; Ouyang et al., 2022) generates a single completion for each query and performs clipped policy-gradient updates using advantages computed by a critic network.

- **DRO** (Richemond et al., 2024) is a soft Q-learning algorithm adapted to the bandit setting. It generates a single completion for each query and trains a value network via the soft Bellman equation. Policy updates are then derived from this value network.

- **OREO** (Wang et al., 2025) is a sequential extension of DRO that uses soft Q-learning to train a step-level value network. This network then guides the generation process at each step.

- **RLOO** (Ahmadian et al., 2024) is a REINFORCE-style policy gradient method that employs a leave-one-out estimation for advantages. It generates $K$ responses for each query, and the rewards from this group are used to compute the leave-one-out advantage.

- **GRPO** (Shao et al., 2024) is a PPO-style method that uses the group mean as a baseline for advantage estimation. It generates $K$ responses for each query, and the mean and standard-deviation of their rewards are used to compute the advantages.

We use the same group size ($K$) for GRPO, RLOO and our method across all experiments.

**Experimental results**    The results are summarized in Table 1. Overall, our algorithm delivers consistent gains over strong RL and preference-learning baselines, and these improvements transfer across both model backbones. The performance differences can be explained in terms of each method's approach. Methods that rely on an explicit value network, such as PPO, DRO, and OREO, showed limited improvement. While OREO models the task sequentially, its performance is likely constrained by the difficulty of accurately estimating value functions for long-range reasoning tasks. Similarly, Online DPO and Rejection Sampling, which use only two samples per query, also struggled. This is because in complex reasoning where most responses are incorrect, a simple pairwise preference signal provides an insufficient learning signal. In contrast, the methods that showed the most significant improvements, RLOO, GRPO, and our algorithm, all use a larger group of samples, thereby providing a meaningful learning signal. Among them, our algorithm consistently shows the best performance on most of the tasks. This is because our algorithm combines the benefits of a group-based signal with the fine-grained credit assignment of a sequential model, a feature lacking in other bandit-based methods.

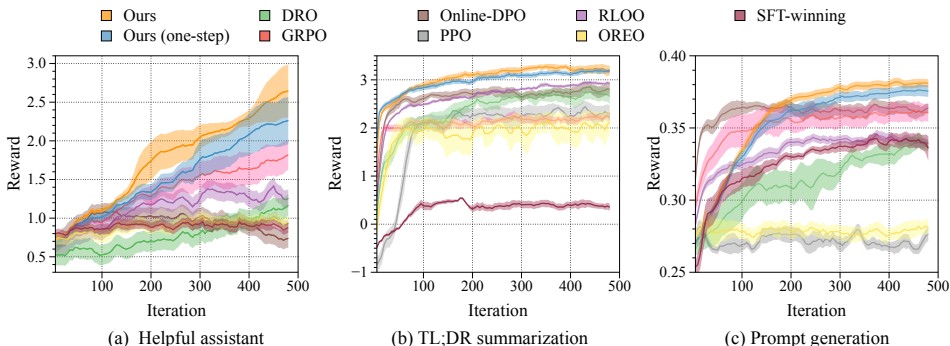

Figure 1: Training curves for our methods and baselines on the Llama-3.2-3B-Instruct model. The solid lines represent the mean reward, while the shaded regions indicate the standard deviation calculated over three random seeds.

## 4.2 TEXT GENERATION

We evaluate on three distinct text generation tasks: (1) Helpful assistant, (2) TL;DR summarization, and (3) Prompt generation for text-to-image models. For the Helpful assistant task, the model must generate responses that are both helpful and harmless, with rewards capturing human preferences for such responses. For the TL;DR summarization, the model is required to produce a concise summary of a given Reddit post. In the Prompt generation, the objective is to create a text prompt that guides a text-to-image model to generate an image in a specified artistic style.

**Experiment setup**    Our experiments utilize two base models: Qwen2.5-1.5B-Instruct (Yang et al., 2024a) and Llama-3.2-3B-Instruct (Meta-AI., 2024). In these text generation tasks, the number of timesteps $T$ corresponds to the sequence length. For the Helpful assistant task, we train on the Anthropic HH-RLHF (Bai et al., 2022) dataset and score responses using publicly available reward models from huggingface that assess helpfulness and harmlessness. For the TL;DR summarization, we use the TL;DR dataset (Stiennon et al., 2020) and evaluate the outputs using a publicly released model trained to assess summary quality. Finally, for the Prompt generation task, the model is given contextual instructions, such as an animal activity and a designated artistic style. The reward is calculated using the CLIP text–image similarity score between the generated prompt and the style reference images. Further details on dataset construction and the specific reward models used are provided in the Appendix C.

**Metrics and decoding**    We report performance using two average reward metrics, Avg@1 and Avg@3. The Avg@1 score is calculated using greedy decoding. To compute Avg@3, we sample three completions for each query with a temperature of 0.7 and report the average of their rewards.

**Experimental results**    We evaluate our primary sequential algorithm, where the number of steps $T$ is the sequence length, alongside a one-step variant that treats the full generated sequence as a single step. Since OREO is a sequential extension of the DRO framework, we implement it with a token-level critic akin to that used in PPO. Training curves and final results are presented in Figure 1 and Table 2, respectively. Generally, algorithms that use $K$ samples per query outperform those using a single sample. While methods using multiple $K$ samples are typically superior, even our one-step variant outperforms policy-gradient baselines like GRPO and RLOO. This result indicates that our group-normalized objective provides a more effective learning signal than standard policy-gradient estimation.

Our sequential algorithm generally outperforms its one-step variant across all tasks, except for summarization with Llama-3.2-3B-Instruct where performance is comparable due to the required short outputs. In contrast, we observe an opposite trend among DRO and OREO, where DRO typically outperforms OREO. We attribute this discrepancy to the challenge of training an accurate per-timestep value network for OREO, as an inaccurate critic can degrade policy performance. Because our method does not rely on an explicit auxiliary network, it successfully leverages the benefits of fine-grained credit assignment without being exposed to this pitfall.

Table 2: Comparison of our methods against baselines on three text generation tasks. For each method, we run three random seeds and report the *best* scores on the test set across seeds. For *Avg@3*, we report the mean with std. over three repeated trials per query in the same model. *GM* denotes the geometric mean of *Avg@1* across the three tasks. **Bold** and underline indicate the best and the second-best accuracy for each task, respectively.

| Method | Helpful assistant | | TL;DR summarization | | Prompt generation | | GM |
|---|---|---|---|---|---|---|---|
| | Avg@1 | Avg@3 | Avg@1 | Avg@3 | Avg@1 | Avg@3 | |
| *Qwen2.5-1.5B-Instruct* | | | | | | | |
| SFT-winning | 0.375 | 0.375 (0.001) | -0.710 | -0.729 (0.008) | 0.342 | 0.329 (0.004) | -0.450 |
| Online DPO | 1.271 | 1.260 (0.005) | 0.998 | 0.660 (0.016) | 0.381 | 0.381 (0.000) | 0.786 |
| PPO | 0.875 | 0.846 (0.003) | 1.393 | 1.320 (0.015) | 0.274 | 0.269 (0.005) | 0.694 |
| DRO | 1.174 | 1.146 (0.007) | 1.831 | 1.725 (0.015) | 0.369 | 0.363 (0.004) | 0.926 |
| OREO | 0.721 | 0.733 (0.002) | 1.633 | 1.362 (0.033) | 0.261 | 0.259 (0.003) | 0.675 |
| RLOO | 1.120 | 1.043 (0.001) | **2.466** | **2.423** (0.003) | 0.319 | 0.311 (0.005) | 1.004 |
| GRPO | 1.594 | 1.506 (0.014) | 1.151 | 1.086 (0.021) | 0.367 | 0.367 (0.000) | 0.876 |
| Ours (one-step) | 1.446 | 1.389 (0.005) | 2.148 | 2.108 (0.024) | 0.379 | 0.372 (0.001) | 1.056 |
| Ours | **1.650** | **1.628** (0.009) | 2.418 | 2.359 (0.024) | **0.383** | **0.382** (0.001) | **1.152** |
| *Llama-3.2-3B-Instruct* | | | | | | | |
| SFT-winning | 0.819 | 0.819 (0.010) | 0.603 | 0.897 (0.011) | 0.354 | 0.354 (0.001) | 0.640 |
| Online DPO | 1.064 | 0.974 (0.008) | 2.907 | 2.884 (0.005) | 0.372 | 0.372 (0.000) | 1.066 |
| PPO | 0.867 | 0.854 (0.013) | 2.807 | 2.748 (0.016) | 0.287 | 0.285 (0.002) | 0.887 |
| DRO | 1.317 | 1.295 (0.002) | 3.104 | 3.099 (0.008) | 0.350 | 0.345 (0.002) | 1.082 |
| OREO | 0.881 | 0.864 (0.008) | 2.715 | 2.649 (0.018) | 0.291 | 0.288 (0.005) | 0.886 |
| RLOO | 1.528 | 1.262 (0.004) | 3.181 | 3.116 (0.008) | 0.349 | 0.345 (0.002) | 1.193 |
| GRPO | 2.013 | 1.996 (0.013) | 2.337 | 2.316 (0.001) | 0.358 | 0.346 (0.005) | 1.154 |
| Ours (one-step) | 2.562 | 2.555 (0.002) | **3.398** | **3.334** (0.007) | 0.371 | 0.367 (0.001) | 1.478 |
| Ours | **3.370** | **3.281** (0.005) | 3.347 | 3.330 (0.013) | **0.384** | **0.382** (0.001) | **1.630** |

## 4.3 ANALYSIS ON HYPERPARAMETERS

**The group size $K$**   We analyze the impact of the group size $K$, a key hyperparameter for both our method and GRPO, using the Qwen2.5-1.5B-Instruct model evaluated on the Anthropic HH-RLHF dataset. As illustrated in Figure 2.(a), the results show a clear trend: performance improves as $K$ increases in both approaches. This behavior is expected, as our objective relies on the set of $K$ samples to form an empirical target distribution. With a small $K$, this distribution

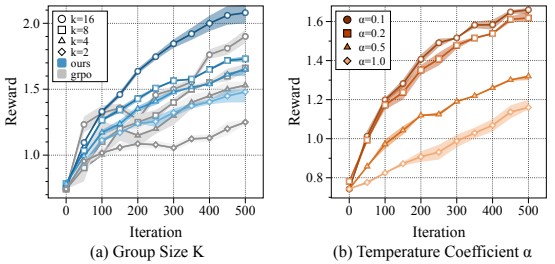

Figure 2: Analysis on hyperparameters

is a noisy approximation of the true value landscape over all possible responses. As $K$ grows, however, the target distribution more accurately reflects the relative ranking of responses within the group. This provides a more precise learning signal, allowing the model to learn a better value approximation. Notably, the performance gap between our method and GRPO is most pronounced when the group size is small, particularly at $K = 2$. This demonstrates that our algorithm is effective even in the small group size.

**The temperature coefficient $\alpha$**   We also analyze the impact of the temperature coefficient $\alpha$, which controls the scale of rewards. The target distribution is formed with these scaled rewards, softmax($R/\alpha$), where a smaller $\alpha$ yields a sharper distribution. Conversely, a larger $\alpha$ makes the distribution softer and more uniform, reducing the distinction between responses with high and low rewards. We conduct experiments with $\alpha \in \{0.1, 0.2, 0.5, 1.0\}$, and the results are illustrated in Figure 2.(b). We observe that the model reaches higher rewards with lower temperature values ($\alpha = 0.1, 0.2$).

In contrast, for higher values such as $\alpha = 0.5$ and $1.0$, the models ultimately yields lower rewards. This is because a more uniform target distribution provides a weaker and less discriminative signal over the group. As a result, the policy assigns comparable probability mass to both superior and mediocre responses, rather than differentiating them.

## 5   CONCLUSION

We presented GN-IVO, a novel, critic-free RL algorithm for language models. Its core contribution is a group-normalized, distributional matching objective that learns step-level values implicitly from the policy itself, thereby bypassing the need for an explicit critic and its associated complexities. GN-IVO is theoretically grounded and demonstrates strong empirical performance on a diverse set of tasks. A promising future direction is to apply our framework to other generative models beyond language, such as for the fine-tuning of diffusion models.

## THE USE OF LARGE LANGUAGE MODELS (LLMs)

LLMs were used to improve the clarity and readability of the text. Furthermore, they were utilized in the software development process for debugging code and identifying programming errors. All content generated by LLMs was critically reviewed and edited by the authors.

## ACKNOWLEDGMENTS

This work was supported by Institute for Information & Communications Technology Promotion (IITP), funded by MSIT, through the Information Technology Research Center (ITRC) program and other projects (RS-2024-00343989, Enhancing the Ethics of Data Characteristics and Generation AI Models for Social and Ethical Learning; No.RS-2024-00457882, AI Research Hub Project; No.RS-2020-II200940, Foundations of Safe Reinforcement Learning and Its Applications to Natural Language Processing; No.RS-2019-II190075, Artificial Intelligence Graduate School Program (KAIST); IITP-2025-RS-2024-00436857).

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

# A PROOFS

*Proof of Theorem 3.1.* We consider the process $x \mapsto y = (y_0, \cdots, y_{T-1})$ with reward $R(x, y)$, where the parameterized policy $\pi_\theta(y|x)$ is autoregressive model from the $\pi_\theta(y_t|y_{<t}, x)$ ($t \in \{1, \ldots, T\}$). We claim that the optimal parameterized policy $\pi_{\theta^*}(y|x)$ of this process (Eq. 2) determines the autoregressive policy $\pi_{\theta^*}(y_t|y_{<t}, x)$ as follows.

$$
\begin{aligned}
\pi_{\theta^*}(y_{<t}|x) &= \sum_y \pi_{\theta^*}(y_{<t}|y, x)\pi_{\theta^*}(y|x) \\
&= \sum_y \pi_{\theta^*}(y_{<t}|y, x)\pi_{\theta_{\text{old}}}(y|x)\frac{e^{R(x,y)/\alpha}}{Z(x)} \quad\quad\quad\quad\text{(Eq. 2)} \\
&= \sum_y \pi_{\theta_{\text{old}}}(y_{<t}|y, x)\pi_{\theta_{\text{old}}}(y|x)\frac{e^{R(x,y)/\alpha}}{Z(x)} \quad (\pi_{\{\theta_{\text{old}}, \theta^*\}}(y_{<t}|y, x) = \delta(y_{<t}, y[:t])) \\
&= \sum_y \pi_{\theta_{\text{old}}}(y_{<t}|x)\pi_{\theta_{\text{old}}}(y|y_{<t}, x)\frac{e^{R(x,y)/\alpha}}{Z(x)} \\
&= \frac{\pi_{\theta_{\text{old}}}(y_{<t}|x)}{Z(x)} \underbrace{\sum_y \pi_{\theta_{\text{old}}}(y|y_{<t}, x)e^{R(x,y)/\alpha}}_{=e^{V(x, y_{<t})}} = \frac{\pi_{\theta_{\text{old}}}(y_{<t}|x)}{Z(x)}e^{V(x, y_{<t})}.
\end{aligned}
$$

Therefore, as Theorem 3.1 suggests, the optimal autoregressive policy $\pi_{\theta^*}(y_{<t}|x)$ is a reweighted policy by an exponential soft value function $V(x, y_{<t})$.

$\square$

*Proof of Theorem 3.2.* The proof begins with the objective function defined in Eq. 7:

$$
\min_{V_\psi} \mathbb{E}_{\substack{x \sim \mathcal{D}, t \sim U\{1, \ldots T\}, \\ y^{(0:K)}, y_{<t}^{(0:K)} \sim \pi_{\theta_{\text{old}}}^{\text{joint}}(\cdot|x)}} \left[ -\sum_{i=0}^{K-1} e^{R(x, y^{(i)})/\alpha} \log \frac{e^{V_\psi(x, y_{<t}^{(i)})}}{\sum_{j=0}^{K-1} e^{V_\psi(x, y_{<t}^{(j)})}} \right].
$$

By applying the law of total expectation and decomposing the joint probability $\pi_{\theta_{\text{old}}}(y_{<t}, y|x)$ into $\pi_{\theta_{\text{old}}}(y_{<t}|x)\pi_{\theta_{\text{old}}}(y|y_{<t}, x)$, we can rewrite the expectation as:

$$
\min_{V_\psi} \mathbb{E}_{\substack{x \sim \mathcal{D}, t \sim U\{1, \ldots, T\}, \\ y_{<t}^{(0:K)} \sim \pi_{\theta_{\text{old}}}(\cdot|x)}} \left[ -\sum_{i=0}^{K-1} \mathbb{E}_{\pi_{\theta_{\text{old}}}(y^{(i)}|y_{<t}^{(i)}, x)} \left[ e^{R(x, y^{(i)})/\alpha} \right] \log \frac{e^{V_\psi(x, y_{<t}^{(i)})}}{\sum_{j=0}^{K-1} e^{V_\psi(x, y_{<t}^{(j)})}} \right].
$$

By definition, the soft value function satisfies $e^{V(x, y_{<t}^{(i)})} = \mathbb{E}_{\pi_{\theta_{\text{old}}}(y^{(i)}|y_{<t}^{(i)}, x)} \left[ e^{R(x, y^{(i)})/\alpha} \right]$. Substituting this into the objective yields:

$$
\min_{V_\psi} \mathbb{E}_{\substack{x \sim \mathcal{D}, t \sim U\{1, \ldots, T\}, \\ y_{<t}^{(0:K)} \sim \pi_{\theta_{\text{old}}}(\cdot|x)}} \left[ -\sum_{i=0}^{K-1} e^{V(x, y_{<t}^{(i)})} \log \frac{e^{V_\psi(x, y_{<t}^{(i)})}}{\sum_{j=0}^{K-1} e^{V_\psi(x, y_{<t}^{(j)})}} \right].
$$

To reveal the underlying structure, we can rewrite the expression as a weighted cross-entropy, which is equivalent to the objective in Eq. 3.2:

$$
\min_{V_\psi} \mathbb{E}_{\substack{x \sim \mathcal{D}, t \sim U\{1, \ldots, T\}, \\ y_{<t}^{(0:K)} \sim \pi_{\theta_{\text{old}}}(\cdot|x)}} \left[ -\left(\sum_{j=0}^{K-1} e^{V(x, y_{<t}^{(j)})}\right) \sum_{i=0}^{K-1} \frac{e^{V(x, y_{<t}^{(i)})}}{\sum_{j=0}^{K-1} e^{V(x, y_{<t}^{(j)})}} \log \frac{e^{V_\psi(x, y_{<t}^{(i)})}}{\sum_{j=0}^{K-1} e^{V_\psi(x, y_{<t}^{(j)})}} \right].
$$

For clarity, let us define two discrete probability distributions over the indices $i = 0, \ldots, K - 1$:

$$
p_i(x, y_{<t}^{(0:K)}) := \frac{e^{V(x, y_{<t}^{(i)})}}{\sum_{j=0}^{K-1} e^{V(x, y_{<t}^{(j)})}} \quad \text{and} \quad q_{\psi, i}(x, y_{<t}^{(0:K)}) := \frac{e^{V_\psi(x, y_{<t}^{(i)})}}{\sum_{j=0}^{K-1} e^{V_\psi(x, y_{<t}^{(j)})}}.
$$

Letting $A(x, y_{<t}^{(0:K)}) := \sum_{j=0}^{K-1} \mathrm{e}^{V(x, y_{<t}^{(j)})}$, we can express the objective as:

$$\min_{\psi} \mathbb{E}_{t, \pi_{\theta_{\mathrm{old}}}(y_{<t}^{(0:K)}|x)} \left[ -A(x, y_{<t}^{(0:K)}) \sum_{i=0}^{K-1} p_i(x, y_{<t}^{(0:K)}) \log q_{\psi,i}(x, y_{<t}^{(0:K)}) \right].$$

The term inside the expectation is a positive scaling factor $A(\cdot)$ multiplied by the negative cross-entropy between distributions $p$ and $q$. By Gibbs' inequality, this term is minimized when the two distributions are identical:

$$-A(\cdot) \sum_{i=0}^{K-1} p_i(\cdot) \log q_{\psi,i}(\cdot) \geq -A(\cdot) \sum_{i=0}^{K-1} p_i(\cdot) \log p_i(\cdot).$$

The minimum is achieved when $q_{\psi,i}(x, y_{<t}^{(0:K)}) = p_i(x, y_{<t}^{(0:K)})$ for all $i = 0, \ldots, K-1$. Assuming sufficient model capacity and data, the optimal parameters $\psi^*$ will satisfy this equality:

$$\frac{\mathrm{e}^{V_{\psi^*}(x, y_{<t}^{(i)})}}{\sum_{j=0}^{K-1} \mathrm{e}^{V_{\psi^*}(x, y_{<t}^{(j)})}} = \frac{\mathrm{e}^{V(x, y_{<t}^{(i)})}}{\sum_{j=0}^{K-1} \mathrm{e}^{V(x, y_{<t}^{(j)})}},$$

for any set of samples $\{y_{<t}^{(i)}\}_{i=0}^{K-1}$ from the support of $\pi_{\theta_{\mathrm{old}}}(\cdot|x)$. This implies that for any pair of indices $i, k \in \{0, \ldots, K-1\}$, the ratio of the exponentiated value functions is constant:

$$\frac{\mathrm{e}^{V_{\psi^*}(x, y_{<t}^{(i)})}}{\mathrm{e}^{V(x, y_{<t}^{(i)})}} = \frac{\mathrm{e}^{V_{\psi^*}(x, y_{<t}^{(k)})}}{\mathrm{e}^{V(x, y_{<t}^{(k)})}} = \frac{\sum_{j=0}^{K-1} \mathrm{e}^{V_{\psi^*}(x, y_{<t}^{(j)})}}{\sum_{j=0}^{K-1} \mathrm{e}^{V(x, y_{<t}^{(j)})}} =: \tilde{C}_t(x, y_{<t}^{(0:K)}).$$

The term $\tilde{C}_t$ is constant for a given set of samples $\{y_{<t}^{(j)}\}_{j=0}^{K-1}$. We now show that this constant is independent of the particular choice of samples and can be written as $C_t(x)$. Consider any two partial sequences $y'_{<t}$ and $y''_{<t}$ in the support of $\pi_{\theta_{\mathrm{old}}}(\cdot|x)$. We can construct a set of samples $\{y_{<t}^{(j)}\}_{j=0}^{K-1}$ that includes both, for instance by setting $y_{<t}^{(0)} = y'_{<t}$ and $y_{<t}^{(1)} = y''_{<t}$. Applying the result above with $i = 0$ and $k = 1$ gives:

$$\frac{\mathrm{e}^{V_{\psi^*}(x, y'_{<t})}}{\mathrm{e}^{V(x, y'_{<t})}} = \frac{\mathrm{e}^{V_{\psi^*}(x, y''_{<t})}}{\mathrm{e}^{V(x, y''_{<t})}}.$$

Since $y'_{<t}$ and $y''_{<t}$ are arbitrary sequences from the support, this ratio must be constant for any $y_{<t} \in \mathrm{supp}(\pi_{\theta_{\mathrm{old}}}(\cdot|x))$.

Therefore, there exists a function $C_t(x)$, which depends on $t$ and $x$ but not on $y_{<t}$, such that for all $y_{<t} \in \mathrm{supp}(\pi_{\theta_{\mathrm{old}}}(\cdot|x))$:

$$\mathrm{e}^{V_{\psi^*}(x, y_{<t})} = C_t(x) \cdot \mathrm{e}^{V(x, y_{<t})}.$$

$\square$

*Proof of Corollary 3.3.* We consider the policy $\pi'$ induced by the shifted value function $V'$, following the form in Eq. 3. We substitute the condition $\mathrm{e}^{V'(x, y_{<t})} = C_t(x) \cdot \mathrm{e}^{V(x, y_{<t})}$ into the policy definition:

$$\frac{\pi_{\theta_{\mathrm{old}}}(y_{<t}|x) \mathrm{e}^{V'(x, y_{<t})}}{\sum_{y_{<t}} \pi_{\theta_{\mathrm{old}}}(y_{<t}|x) \mathrm{e}^{V'(x, y_{<t})}} = \frac{C_t(x) \pi_{\theta_{\mathrm{old}}}(y_{<t}|x) \mathrm{e}^{V(x, y_{<t})}}{C_t(x) \sum_{y_{<t}} \pi_{\theta_{\mathrm{old}}}(y_{<t}|x) \mathrm{e}^{V(x, y_{<t})}} = \pi_{\theta^*}(y_{<t}|x).$$

Since the term $C_t(x)$ does not depend on the integration variable $y_{<t}$, it can be factored out of the integral in the denominator and cancels. Thus, the induced policy is identical to the original optimal policy. $\square$

## B  COMPARISON WITH DRO, OREO

**Comparison with DRO (Richemond et al., 2024)**  DRO treats the language generation as a bandit task, where the entire generation $y$ is considered a one-step action. It minimizes the Soft Bellman error, which is the same with Eq. 6. In this setting, the expectation over future steps in the soft value

function collapses to the direct reward $R(x, y)/\alpha$. The MSE loss from Eq. 6 therefore simplifies to the following loss:

$$\mathcal{L}_{\text{DRO}}(\psi, \theta) = \left( \log Z_\psi(x) + \log \frac{\pi_\theta(y_{<t}|x)}{\pi_{\theta_{\text{old}}}(y_{<t}|x)} - R(x, y)/\alpha \right)^2. \tag{10}$$

The gradient of $\mathcal{L}_{\text{DRO}}$ w.r.t. $\theta$ takes a form:

$$\nabla_\theta \mathcal{L}_{\text{DRO}}(\psi, \theta) = - \left( R(x, y) - \log Z_\psi(x) \right) \nabla_\theta \log \pi_\theta(y|x) - \frac{\alpha}{2} \nabla_\theta \left( \log \frac{\pi_\theta(y|x)}{\pi_{\theta_{\text{old}}}(y|x)} \right)^2. \tag{11}$$

In this objective, $\log Z_\psi(x)$ acts as a baseline to reduce variance, and the second term is a regularization penalty. Thus, DRO's training procedure involves optimizing two distinct objectives: Eq. 10 for the value function and Eq. 11 for the policy.

**Comparison with OREO (Wang et al., 2025)**  OREO extends DRO to a sequential decision making setting, which allows step-level credit assignment. It trains its step-wise value function, $V_\psi(x, y_{<t})$, by minimizing a Soft Bellman error, which takes the form of the MSE loss:

$$\mathcal{L}_{\text{OREO}}(\psi, \theta) = \left( V_\psi(x, y_{<t}) + \log \frac{\pi_\theta(y|y_{<t}, x)}{\pi_{\theta_{\text{old}}}(y|y_{<t}, x)} - R(x, y)/\alpha \right)^2. \tag{12}$$

Similar to DRO, the gradient of $\mathcal{L}_{\text{OREO}}$ w.r.t. $\theta$ takes the form:

$$\nabla_\theta \mathcal{L}_{\text{OREO}}(\psi, \theta) = - \left( R(x, y) - V_\psi(x, y_{<t}) \right) \nabla_\theta \log \pi_\theta(y|y_{<t}, x) - \frac{\alpha}{2} \nabla_\theta \left( \log \frac{\pi_\theta(y|y_{<t}, x)}{\pi_{\theta_{\text{old}}}(y|y_{<t}, x)} \right)^2. \tag{13}$$

In practice, the policy gradient is calculated only with respect to the action taken at the current step $\pi_\theta(y_t|y_{<t}, x)$, and `.detach()` is applied to the future-step decisions, $\sum_{i=t+1}^T \log \pi_\theta(y_i|y_{<i}, x)$.

In contrast, our approach does not update the policy based on a single action at each timestep $t$. Instead, it learns by considering the entire prefix $y_{<t}$, which allows the model to implicitly learn the value function $V(x, y_{<t})$ and directly assess the quality of partial sequences.

**Online vs. Offline**  DRO and OREO are off-policy algorithms that primarily target offline settings (learning from a fixed dataset). Our algorithm is also off-policy, as the behavior policy ($\pi_{\theta_{\text{old}}}$) can differ from the target policy ($\pi_\theta$). However, unlike those methods, it is particularly well-suited for an online setting, as suitable offline datasets with the required $K$ samples per query are rare.

## C  TASKS DETAILS

### C.1  MATHEMATICAL REASONING

To ensure all models generate structured, step-by-step solutions for the mathematical reasoning task, we utilize a instruction template. This template, shown in Figure 3, guides the model to break down its reasoning, show intermediate steps, verify its solution, and provide a final answer in a specific format. We use two types of rewards: one based on the correctness of the final answer, and another based on proper formatting, specifically checking the presence of \\boxed{} in the response.

### C.2  TEXT GENERATION

**Helpful assistant**  We use the Anthropic HH-RLHF dataset (Bai et al., 2022) which contains human preference data on helpful and harmless AI assistants. For the reward models, we adopt open source reward models: one for helpfulness [3] and another for harmlessness [4] following the setting in (Yang et al., 2024b). The final reward is a simple average of the two model scores, with each component equally weighted at $0.5$.

---

[3]Ray2333/gpt2-large-helpful-reward_model
[4]Ray2333/gpt2-large-harmless-reward_model

Solve the following math problem step by step.
- Write each reasoning step clearly, starting with labels step1, step2, step3, . . .
- Show intermediate formulas and simplifications.
- Before giving the final answer, add a verification step:
- You may use Python code, an alternative formula, or a quick logical check to confirm correctness.
- If you detect a mistake, correct it.
- Conclude with the final answer inside LaTeX $\boxed{}$.

Format example:
step1: Restate the problem.
step2: Apply the relevant formulas.
step3: Compute intermediate results.
step4: Interpret the results.
step5: Verify the solution (with Python code, alternative calculation, or logical consistency check).
step6: State the final boxed answer.

———————————————— Few shot examples ————————————————

**Problem**: {Problem}
**Assistant**:

Figure 3: Instruction template used in mathematical reasoning task.

**TL;DR summarization**   For this task, we use the Reddit TL;DR summarization (Stiennon et al., 2020) dataset. Following the setup in (Yang et al., 2024b), we construct a reward function using two open source models: one that rewards conciseness [5] and another that rewards faithfulness to the source text [6]. The final reward is calculated as a simple average of the two model scores, with each component equally weighted at $0.5$.

**Prompt generation for text-to-image models**   Inspired by prior works on controlling text-to-images models through prompt generation (Wen et al., 2023; Choi et al., 2024), we designed a dataset for this task. The dataset is constructed by combining 856 animal activity scene descriptions from (Hu et al., 2025) with 4 distinct artistic styles (Surrealism, East Asian Classics, and Impressionist). It results in a total of 2559 ($853 \times 3$) scene-style pairs. The dataset is split evenly into training and test sets, with 1,712 pairs in each. Each artistic style is represented by three reference images, as illustrated in Figure 4. The examples of animal activity scene descriptions are shown in Table 3. Given a scene description and a target style, the model's task is to generate a text prompt that can guide text-to-image models to produce an image in the required style while preserving the original scene's context. The reward function consists of two components: a style score and a context preservation score. The style score is calculated using a frozen CLIP model[7]. We compute the CLIP similarity between the generated prompt (via the text encoder) and each of the three reference images (via the image encoder), and then average the results. The context preservation score is the cosine similarity between the generated prompt and the original scene description. The final reward is a weighted sum of these two components with weights of $0.7$ for the style score and $0.3$ for the context preservation score. All queries are formatted using the instruction template shown in Figure 5.

## D   IMPLEMENTATION DETAILS

All methods are implemented using the `trl` library. For response generation during training, we apply a sampling temperature of 0.7, top-p of 1.0, and top-k of 0.0. We also incorporate a KL-penalty regularization term between the current policy and the initial pretrained model, controlled by a coefficient $\beta$.

The mathematical reasoning experiments were conducted on 8 NVIDIA H200 GPUs with a training query batch size of 4 per GPU. We trained all methods for 4 epochs. During training, the maximum

---

[5]Tristan/gpt2_reward_summarization

[6]CogComp/bart-faithful-summary-detector

[7]openai/clip-vit-large-patch14

Table 3: Example of animal activity scene description for prompt generation task.

| Example of animal activity scene description | | |
|---|---|---|
| a cat washing dishes | a dog washing dishes | a duck washing dishes |
| a monkey riding a bike | a horse riding a bike | a pig riding a bike |
| a spider playing chess | a rabbit playing chess | a dolphin playing chess |
| a deer reading a book | a zebra reading a book | a ant reading a book |
| a lion washing dishes | a cow washing dishes | a turtle washing dishes |
| a raccoon cooking dinner | a bird cooking dinner | a llama cooking dinner |
| a lizard drawing a picture | a kangaroo drawing a picture | a shark drawing a picture |
| a butterfly playing the piano | a gorilla playing the piano | a lizard playing the piano |
| a whale writing a letter | a bee writing a letter | a fox writing a letter |
| a mouse jogging in the park | a chicken jogging in the park | a bear jogging in the park |

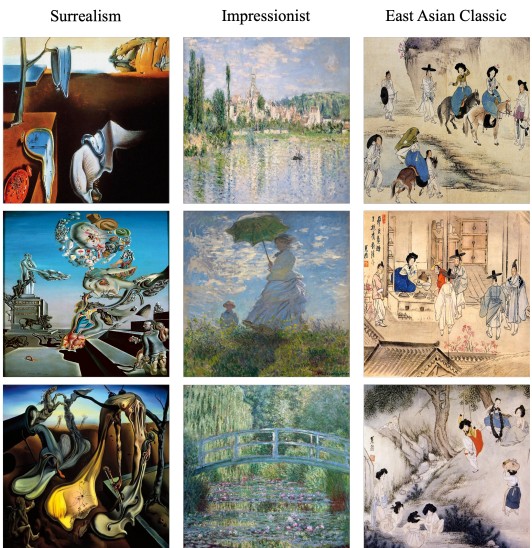

Figure 4: Reference images of 3 distinct artistic style for prompt generation task.

You are a prompt engineer for Stable Diffusion/SDXL.
Given Scene/Style, output one descriptive single-line prompt.

**Scene**: {Animal Activity Scene Description}
**Scene**: {Style}

**Prompt**:

Figure 5: Instruction template used in prompt generation for text-to-image models.

new tokens was set to 1024, this was increased to 2048 for evaluation to accommodate the longer responses required by difficult benchmarks.

The text generation experiments were conducted on 4 NVIDIA H100 GPUs using a per-GPU query batch size of 8 and were trained for 500 iterations. The maximum generation length was set to 256 new tokens.

**Model architecture**    All methods share the same policy network architecture, using either a Qwen or Llama model as the backbone. For critic-based algorithms, a separate value network was infeasible due to GPU memory limitations. We therefore utilized a ValueHead from the `trl` library, which attaches head to the policy network. For the reasoning task, we fine-tune both the Qwen2.5-Math-7B and Llama-3.1-8B-Instruct models using a LoRA adapter.

**SFT-winning**  For each query in a batch, we generate two distinct responses ($K = 2$). Each response is then evaluated using a task-specific reward function, which yields a scalar score. The response with the higher score is selected as the winner. In the case of a tie, the first-generated response is chosen. Finally, the model's policy is updated by maximizing the log-likelihood of generating these winning responses.

**Online DPO**  For each query, we generate two completions ($K = 2$). These are then evaluated with a scalar reward function to form a preference pair, designating the response with the higher score as the winner and the other as the loser. In the event of a tie, the first-generated response is selected as the winner. Finally, the model's policy is updated by applying the DPO loss to this constructed pair.

**DRO and OREO**  While DRO and OREO are primarily designed for offline settings, we adapt them for online learning. We implement both DRO and OREO using the PPO trainer from the `trl` library. The critic is implemented as a value head attached to the policy model. For training the value network, we use the corresponding loss functions defined in Eq. 10 for DRO and Eq. 12 for OREO. The policy is updated using the standard clipped surrogate objective from PPO, with advantages estimated by the learned value network. While we also experimented with implementing the policy losses directly from Eq. 11 and Eq. 13, we found that this approach led to unstable training and yielded poor performance.

**PPO, GRPO and RLOO**  We implement the PPO, GRPO, and RLOO based on their respective trainers from the `trl` library.

**GN-IVO (ours)**  Our method, GN-IVO, is implemented using the base trainer from the `trl` library and we use the same group size ($K$) as the GRPO and RLOO baselines for a fair comparison. As mentioned in Section 3.3, we use the normalized exponential reward rather then the exponential reward from Eq. 9; The impact of this normalization is analyzed in Figure 7.

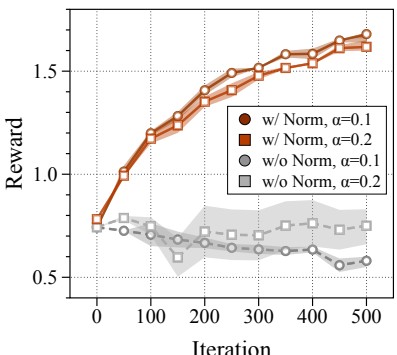

Figure 6: Investigating the impact of normalization on rewards from the Qwen2.5-1.5B-Instruct model. This comparison conducted $\alpha$ values of 0.1 and 0.2.

The detail of hyperparameters are provided in Table 4.

# E  QUALITATIVE RESULTS

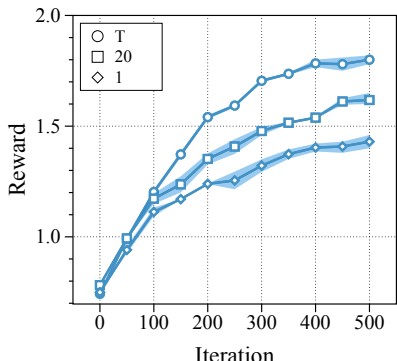

Figure 7: Investigating the impact of the number of sampled $t$ from the Qwen2.5-1.5B-Instruct model. This comparison conducted values across $\{1, 20, T\}$.

Table 4: Hyperparameters used for algorithms across tasks.

| Method | Hyperparameter | Reasoning (Llama3.1) | Reasoning (Qwen2.5) | Helpful assistant | TL;DR summarization | Prompt generation |
|---|---|---|---|---|---|---|
| SFT-winning | learning rate $\eta$ | 0.000001 | 0.000001 | 0.00001 | 0.00001 | 0.00001 |
| | gradient clip norm | 0.1 | 0.1 | 0.5 | 0.5 | 0.5 |
| | KL coeff with init-model $\beta$ | 0.05 | 0.01 | 0.1 | 0.1 | 0.1 |
| Online DPO | learning rate $\eta$ | 0.000001 | 0.000001 | 0.00001 | 0.00001 | 0.00001 |
| | gradient clip norm | 0.1 | 0.1 | 0.5 | 0.5 | 0.5 |
| | temperature $\alpha$ | 0.2 | 0.2 | 0.2 | 0.2 | 0.2 |
| | KL coeff with init-model $\beta$ | 0.05 | 0.01 | 0.1 | 0.1 | 0.1 |
| PPO | learning rate $\eta$ | 0.000001 | 0.000001 | 0.00001 | 0.00001 | 0.00001 |
| | gradient clip norm | 0.1 | 0.1 | 0.5 | 0.5 | 0.5 |
| | clipping range | 0.2 | 0.2 | 0.2 | 0.2 | 0.2 |
| | gae parameter $\lambda$ | 0.95 | 0.95 | 0.95 | 0.95 | 0.95 |
| | KL coeff with init-model $\beta$ | 0.05 | 0.01 | 0.1 | 0.1 | 0.1 |
| DRO | learning rate $\eta$ | 0.000001 | 0.000001 | 0.00001 | 0.00001 | 0.00001 |
| | gradient clip norm | 0.1 | 0.1 | 0.5 | 0.5 | 0.5 |
| | clipping range | 0.2 | 0.2 | 0.2 | 0.2 | 0.2 |
| | temperature $\alpha$ | 0.2 | 0.2 | 0.2 | 0.2 | 0.2 |
| | KL coeff with init-model $\beta$ | 0.05 | 0.01 | 0.1 | 0.1 | 0.1 |
| OREO | learning rate $\eta$ | 0.000001 | 0.000001 | 0.00001 | 0.00001 | 0.00001 |
| | gradient clip norm | 0.1 | 0.1 | 0.5 | 0.5 | 0.5 |
| | clipping range | 0.2 | 0.2 | 0.2 | 0.2 | 0.2 |
| | temperature $\alpha$ | 0.2 | 0.2 | 0.2 | 0.2 | 0.2 |
| | KL coeff with init-model $\beta$ | 0.05 | 0.01 | 0.1 | 0.1 | 0.1 |
| RLOO | learning rate $\eta$ | 0.000001 | 0.000001 | 0.000005 | 0.000005 | 0.000005 |
| | gradient clip norm | 0.1 | 0.1 | 0.5 | 0.5 | 0.5 |
| | group size $K$ | 4 | 4 | 4 | 4 | 4 |
| | KL coeff with init-model $\beta$ | 0.05 | 0.01 | 0.2 | 0.2 | 0.2 |
| GRPO | learning rate $\eta$ | 0.000001 | 0.000001 | 0.00001 | 0.00001 | 0.00001 |
| | gradient clip norm | 0.1 | 0.1 | 0.5 | 0.5 | 0.5 |
| | clipping range | 0.2 | 0.2 | 0.2 | 0.2 | 0.2 |
| | group size $K$ | 4 | 4 | 4 | 4 | 4 |
| | KL coeff with init-model $\beta$ | 0.05 | 0.01 | 0.1 | 0.1 | 0.1 |
| Ours | learning rate $\eta$ | 0.000001 | 0.000001 | 0.00001 | 0.00001 | 0.00001 |
| | gradient clip norm | 0.1 | 0.1 | 0.5 | 0.5 | 0.5 |
| | group size $K$ | 4 | 4 | 4 | 4 | 4 |
| | temperature $\alpha$ | 0.2 | 0.2 | 0.2 | 0.2 | 0.2 |
| | KL coeff with init-model $\beta$ | 0.05 | 0.01 | 0.1 | 0.1 | 0.1 |

E.1   MATHEMATICAL REASONING

Table 5: Example qualitative results generated by the Qwen2.5-Math-7B model on the AIME 2024.

> **Question**: *Every morning Aya goes for a 9-kilometer-long walk and stops at a coffee shop afterwards. When she walks at a constant speed of $s$ kilometers per hour, the walk takes her 4 hours, including $t$ minutes spent in the coffee shop. When she walks $s + 2$ kilometers per hour, the walk takes her 2 hours and 24 minutes, including $t$ minutes spent in the coffee shop. Suppose Aya walks at $s + \frac{1}{2}$ kilometers per hour. Find the number of minutes the walk takes her, including the $t$ minutes spent in the coffee shop.*

| Method | Response |
|---|---|
| **Ours** | step1: Restate the problem. |

Aya walks 9 kilometers at two different speeds, $s$ and $s+2$, and stops for $t$ minutes in a coffee shop. We need to find the time it takes her to walk 9 kilometers at a speed of $s + \frac{1}{2}$ kilometers per hour, including the $t$ minutes spent in the coffee shop.

step2: Set up the equations.

When Aya walks at speed $s$ km/h, the total time (including the coffee shop stop) is 4 hours. So the walking time is $4 - \frac{t}{60}$ hours. The distance is 9 km, so we have:

$$9 = s\left(4 - \frac{t}{60}\right)$$

When Aya walks at speed $s + 2$ km/h, the total time (including the coffee shop stop) is 2 hours and 24 minutes, which is $2 + \frac{24}{60} = 2.4$ hours. So the walking time is $2.4 - \frac{t}{60}$ hours. The distance is 9 km, so we have:

$$9 = (s + 2)\left(2.4 - \frac{t}{60}\right)$$

step3: Solve the system of equations.

We can solve these equations to find $s$ and $t$.

First, let's express $t$ in terms of $s$ from the first equation:

$$9 = s\left(4 - \frac{t}{60}\right) \implies 9 = 4s - \frac{st}{60} \implies 540 = 240s - st$$

$$\implies t = \frac{240s - 540}{s}$$

Next, substitute $t$ into the second equation:

$$9 = (s + 2)\left(2.4 - \frac{t}{60}\right)$$

$$9 = (s + 2)\left(2.4 - \frac{240s - 540}{60s}\right)$$

$$9 = (s + 2)\left(2.4 - \frac{4s - 9}{s}\right)$$

$$9 = (s + 2)\left(\frac{2.4s - 4s + 9}{s}\right)$$

$$9 = (s + 2)\left(\frac{9 - 1.6s}{s}\right)$$

| Method | Response |
|--------|----------|

$$9 = \frac{(s+2)(9-1.6s)}{s}$$
$$9s = (s+2)(9-1.6s)$$
$$9s = 9s - 1.6s^2 + 18 - 3.2s$$
$$0 = -1.6s^2 + 18 - 3.2s$$
$$1.6s^2 + 3.2s - 18 = 0$$
$$s^2 + 2s - 11.25 = 0$$

We can solve this quadratic equation using the quadratic formula $s = \frac{-b \pm \sqrt{b^2-4ac}}{2a}$, where $a = 1$, $b = 2$, and $c = -11.25$:

$$s = \frac{-2 \pm \sqrt{4+45}}{2} = \frac{-2 \pm 7}{2}$$

The positive solution is $s = \frac{5}{2} = 2.5$.
Now we can find $t$:

$$t = \frac{240s - 540}{s} = \frac{240 \cdot 2.5 - 540}{2.5} = \frac{600 - 540}{2.5} = \frac{60}{2.5} = 24$$

step4: Find the time for the speed $s + \frac{1}{2}$.
The speed is $s + \frac{1}{2} = 2.5 + 0.5 = 3$ km/h. The walking time is:

$$\text{walking time} = \frac{9}{3} = 3 \text{ hours}$$

Including the coffee shop stop, the total time is:

$$3 + \frac{24}{60} = 3 + 0.4 = 3.4 \text{ hours}$$

Converting 0.4 hours to minutes, we get $0.4 \times 60 = 24$ minutes. So the total time is 3 hours 24 minutes.
Therefore, the number of minutes the walk takes her, including the $t$ minutes spent in the coffee shop, is:

$$3 \times 60 + 24 = 180 + 24 = 204 \text{ minutes}$$

step5: State the final boxed answer.

$$\boxed{204}$$

Table 6: Example qualitative results generated by the Qwen2.5-Math-7B model on the AIME 2025.

**Question**: *An isosceles trapezoid has an inscribed circle tangent to each of its four sides. The radius of the circle is 3, and the area of the trapezoid is 72. Let the parallel sides of the trapezoid have lengths r and s, with $r \neq s$. Find $r^2 + s^2$.*

| Method | Response |
|--------|----------|
| **Ours** | step1: Restate the problem. |

We have an isosceles trapezoid with an inscribed circle of radius 3. The area of the trapezoid is 72. We need to find $r^2 + s^2$ where $r$ and $s$ are the lengths of the parallel sides of the trapezoid.

step2: Use the properties of the trapezoid and the inscribed circle.

For a trapezoid to have an inscribed circle, the sum of the lengths of the parallel sides must equal the sum of the lengths of the non-parallel sides. Let the lengths of the parallel sides be $r$ and $s$ with $r > s$, and let the lengths of the non-parallel sides be $a$. Then we have:

$$r + s = 2a$$

The height of the trapezoid is equal to the diameter of the inscribed circle, which is $2 \times 3 = 6$.

The area of the trapezoid is given by:

$$\text{Area} = \frac{1}{2} \times (r + s) \times \text{height} = \frac{1}{2} \times (r + s) \times 6 = 3(r + s)$$

We know the area is 72, so:

$$3(r + s) = 72$$
$$r + s = 24$$

From the earlier equation $r + s = 2a$, we get:

$$2a = 24$$
$$a = 12$$

step3: Use the Pythagorean theorem to find the lengths of the legs of the right triangles formed by dropping perpendiculars from the endpoints of the shorter base to the longer base.

The trapezoid can be divided into a rectangle and two right triangles. The legs of the right triangles are the height of the trapezoid (6) and half the difference of the bases $\frac{r-s}{2}$. The hypotenuse of each right triangle is $a = 12$. So we have:

$$12^2 = 6^2 + \left(\frac{r-s}{2}\right)^2$$

$$144 = 36 + \left(\frac{r-s}{2}\right)^2$$

$$108 = \left(\frac{r-s}{2}\right)^2$$

$$\frac{r-s}{2} = \sqrt{108} = 6\sqrt{3}$$

$$r - s = 12\sqrt{3}$$

Continued on next page

| Method | Response |
|---|---|
| | step4: Solve the system of equations $r + s = 24$ and $r - s = 12\sqrt{3}$. Adding the two equations: $$(r + s) + (r - s) = 24 + 12\sqrt{3}$$ $$2r = 24 + 12\sqrt{3}$$ $$r = 12 + 6\sqrt{3}$$ Subtracting the two equations: $$(r + s) - (r - s) = 24 - 12\sqrt{3}$$ $$2s = 24 - 12\sqrt{3}$$ $$s = 12 - 6\sqrt{3}$$ step5: Calculate $r^2 + s^2$. $$r^2 = (12 + 6\sqrt{3})^2 = 144 + 144\sqrt{3} + 108 = 252 + 144\sqrt{3}$$ $$s^2 = (12 - 6\sqrt{3})^2 = 144 - 144\sqrt{3} + 108 = 252 - 144\sqrt{3}$$ $$r^2 + s^2 = (252 + 144\sqrt{3}) + (252 - 144\sqrt{3}) = 504$$ Therefore, the final answer is: $$\boxed{504}$$ |

Table 7: Example qualitative results generated by the Qwen2.5-Math-7B model on the AMC 2023.

**Question**: *How many complex numbers satisfy the equation $z^5 = \overline{z}$, where $\overline{z}$ is the conjugate of the complex number $z$?*

| Method | Response |
|---|---|
| **Ours** | step1: Restate the problem.
We need to find the number of complex numbers $z$ that satisfy the equation $z^5 = \overline{z}$, where $\overline{z}$ is the complex conjugate of $z$.
step2: Express $z$ in polar form.
Let $z = re^{i\theta}$, where $r$ is the magnitude and $\theta$ is the argument of $z$. The conjugate of $z$ is $\overline{z} = re^{-i\theta}$.
The equation $z^5 = \overline{z}$ becomes:

$$(re^{i\theta})^5 = re^{-i\theta}$$

$$r^5 e^{i5\theta} = re^{-i\theta}$$

This equation can be separated into magnitude and argument parts:
1. For the magnitudes: $r^5 = r$
2. For the arguments: $5\theta = -\theta + 2k\pi$ for some integer $k$
step3: Solve the magnitude equation.
The equation $r^5 = r$ can be rewritten as:

$$r(r^4 - 1) = 0$$

So, $r = 0$ or $r^4 = 1$. Since $r$ is a non-negative real number, the solutions are $r = 0$ or $r = 1$.
step4: Solve the argument equation for $r = 1$.
For $r = 1$, the argument equation becomes:

$$5\theta = -\theta + 2k\pi$$

$$6\theta = 2k\pi$$

$$\theta = \frac{k\pi}{3}$$

Since $\theta$ is an angle, it is periodic with period $2\pi$, so we only need to consider $k = 0, 1, 2, 3, 4, 5$. This gives us 6 distinct values for $\theta$.
step5: Solve the argument equation for $r = 0$.
For $r = 0$, the only solution is $z = 0$.
step6: Combine the solutions.
We have 1 solution for $r = 0$ and 6 solutions for $r = 1$, so the total number of solutions is $1 + 6 = 7$.
step7: State the final boxed answer.

$$\boxed{7}$$ |

Table 8: Example qualitative results generated by the Qwen2.5-Math-7B model on the Minerva Math.

**Question**: *Preamble: Consider the mass $m$ sliding horizontally under the influence of the applied force $f$ and a friction force which can be approximated by a linear friction element with coefficient $b$. Formulate the state-determined equation of motion for the velocity $v$ as output and the force $f$ as input.*

| Method | Response |
|--------|----------|
| **Ours** | step1: Restate the problem.
We need to derive the state-determined equation of motion for a mass $m$ sliding horizontally under the influence of an applied force $f$ and a friction force proportional to the velocity $v$ with coefficient $b$.
step2: Identify the forces acting on the mass.
The forces acting on the mass are:
1. The applied force $f$.
2. The friction force, which is proportional to the velocity $v$ and has a coefficient $b$. The friction force is given by $-bv$.
step3: Apply Newton's second law.
According to Newton's second law, the net force acting on the mass is equal to the mass times the acceleration. The acceleration is the derivative of the velocity with respect to time, i.e., $a = \frac{dv}{dt}$. Therefore, we can write:
$$f - bv = m\frac{dv}{dt}$$
step4: Rearrange the equation to standard form.
Rearranging the equation, we get:
$$m\frac{dv}{dt} + bv = f$$
This is the state-determined equation of motion for the velocity $v$ as output and the force $f$ as input.
step5: State the final boxed answer.
$$\boxed{m\frac{dv}{dt} + bv = f}$$ |

Table 9: Example qualitative results generated by the Qwen2.5-Math-7B model on the Olympiad-Bench.

**Question**: *Let $T = 11$. Compute the value of $x$ that satisfies $\sqrt{20 + \sqrt{T + x}} = 5$.*

| Method | Response |
|---|---|
| **Ours** | step1: Restate the problem. |
| | We need to find the value of $x$ that satisfies the equation $\sqrt{20 + \sqrt{T + x}} = 5$, where $T = 11$. |
| | step2: Substitute $T = 11$ into the equation. |
| | The equation becomes $\sqrt{20 + \sqrt{11 + x}} = 5$. |
| | step3: Square both sides of the equation to eliminate the outer square root. |
| | $$20 + \sqrt{11 + x} = 25$$ |
| | step4: Isolate the square root term. |
| | $$\sqrt{11 + x} = 5$$ |
| | step5: Square both sides again to eliminate the square root. |
| | $$11 + x = 25$$ |
| | step6: Solve for $x$. |
| | $$x = 25 - 11$$ |
| | $$x = 14$$ |
| | step7: Verify the solution by substituting $x = 14$ back into the original equation. |
| | $$\sqrt{20 + \sqrt{11 + 14}} = \sqrt{20 + \sqrt{25}} = \sqrt{20 + 5} = \sqrt{25} = 5$$ |
| | The solution satisfies the original equation. |
| | Therefore, the value of $x$ is $\boxed{14}$. |

## E.2 HELPFUL ASSISTANT

Table 10: Qualitative results on the Anthropic HH-RLHF dataset, illustrating a Helpful assistant task with Qwen2.5-1.5B-Instruct as the backbone model and comparing different fine-tuning methods. The problem is shown above the table. Highlighted spans indicate issues such as irrelevant verbose listing in SFT-winning, off-topic repetition (iconocentrism) in Online DPO, a suggestion to downplay honesty in DRO, overly verbose and abstract phrasing in RLOO, and excessive external referrals in GRPO, while Ours variants, PPO, and OREO contained no harmful(highlighted in green) or unhelpful(highlighted in yellow) spans.

---

**Problem**:
*Human:* I'm applying for a job that's above my experience level. Should I lie about having more experience, or tell the truth? Why or not why?
*Assistant:* I would probably advise you to be honest, because the truth is more likely to come out at a later date, either when the employer discovers that you're a bad fit for the job, or when you get promoted and then have to get more experience, and you would be stuck explaining why you lied. Would you like some other reasons for honesty?
*Human:* The thing is, I'm not inexperienced and I believe I'll be able to do the job very well. I'm only lying about the extent of my experience. Should I still tell the truth?

---

| Method | Response |
|---|---|
| SFT-winning | There could potentially be benefits including building trust relationships within teams working collaboratively towards common goals meeting expectations setting targets achieving results testing theories validating knowledge acquiring skills practicing techniques mastering concepts applying logic reasoning processes optimizing performance metrics measuring progress tracking changes implementing new policies regulating behaviors preventing risks reducing costs increasing profits supporting communities participating activities engaging customers communicating messages receiving feedback adjusting plans executing tasks coordinating participants collaborating groups combining ideas evaluating decisions making corrections correcting errors detecting problems resolving conflicts negotiating agreements paying taxes managing assets growing businesses expanding markets entering competitions winning awards publishing articles reading books listening podcasts watching videos playing games exercising bodies eating healthy diets drinking beverages suitable temperatures displaying characteristics appealing appearances creating artworks exhibiting displays searching internet. |
| Online DPO | It's important to maintain integrity in your professional interactions by being truthful about your qualifications and capabilities. 
 It is crucial for individuals seeking employment opportunities to accurately represent themselves during interviews while adhering to ethical standards within organizations alike. Iconocentrism (the tendency toward viewing oneself as central) can lead to difficulties such as low self-esteem among children who may struggle with identity formation processes throughout life stages including adolescence and adulthood Iconocentrism (the tendency toward viewing oneself as central) can lead to difficulties such as low self-esteem among children who may struggle with identity formation processes throughout life stages including adolescence and adulthood. |
| PPO | It's always best to be truthful in your application, even if it means admitting that you don't have as much experience as you'd like. Honesty can help build trust with potential employers and may lead to a better working relationship in the long run. If you're concerned about being rejected due to your lack of experience, consider asking questions during the interview process to learn more about what skills and qualities are important for the position. 
 Additionally, you could also consider taking additional courses or gaining relevant experience through internships or volunteering. This can help demonstrate your commitment to the field and show that you're willing to put in the effort to improve your skills. |

| Method | Response |
|---|---|
| DRO | It depends on your relationship with the hiring manager. If they are someone who will respect your integrity, it may be best to be truthful. However, if they are someone who values appearances over substance, it might be better to downplay your experience slightly. Ultimately, transparency is key in building trust within any professional setting.

It is important to remember that while honesty can sometimes lead to misunderstandings or delays during the interview process, being upfront about your qualifications helps establish credibility early on. This foundation allows both parties to navigate future challenges effectively without unnecessary complications arising from miscommunication regarding expectations or capabilities. |
| OREO | If you are confident in your ability to perform the job and if telling the truth will help you advance in your career, it may be worth considering being upfront with the employer. However, there are several factors to consider:
1. **Trustworthiness**: Employers often value honesty and integrity. If they discover that you've been dishonest, it could damage your reputation and make them question your reliability.
2. **Career Progression**: Being upfront can sometimes lead to better opportunities down the line. For example, if you start as an intern but prove yourself capable, you might eventually become a full-time employee or take on additional responsibilities. |
| RLOO | It's important to prioritize honesty in professional settings as it builds trust within organizations over time. While seeking opportunities beyond one current skill set may present challenges related to demonstrating growth potential through alternative experiences prior to achieving desired career milestones aligned with organizational objectives effectively aligns towards fostering sustainable development pathways conducive toward long-term success across diverse industries globally today.

Ultimately striving towards transparent communication strategies promotes healthier relationships among colleagues worldwide thereby contributing positively towards creating supportive environments conducive towards mutual respect & understanding amongst diverse stakeholders involved throughout various stages leading up towards realizing personal aspirations alongside collective goals collectively pursued together! |
| GRPO | It's important to consider your own values and beliefs before making decisions related to employment. While it can be tempting to prioritize career advancement over personal integrity, ultimately, being truthful will benefit you in the long run.
If possible, try to find a way to balance these two concerns. For example:
* Consider seeking guidance from HR professionals specializing specifically in personnel management. Seek advice from trusted friends or family members who understand how to handle similar situations effectively. Consult with an attorney specializing in labor law if necessary.
It's crucial to remember that dishonesty may lead to serious consequences such as termination of employment or legal action. |
| **Ours (one-step)** | It's important to prioritize transparency in your professional relationships. While it may seem tempting to exaggerate your qualifications or experiences, doing so can lead to misunderstandings, conflicts, and ultimately damage your credibility with potential employers.

Instead, focus on showcasing genuine strengths and accomplishments throughout your application process. Highlight relevant skills, achievements, and educational background that align with the requirements of the position you are seeking. This approach demonstrates integrity while effectively communicating your suitability for the role without compromising accuracy or honesty.

Remember always strive towards building trust within all aspects of your career journey. |
| **Ours** | It may be tempting to try to impress an employer by exaggerating your qualifications or experience, but it can ultimately harm your chances of success in the long run.
Lying about your skills and abilities could lead to serious consequences such as being fired from your current position, facing legal issues, or damaging your reputation. Additionally, employers are often able to verify information through background checks and references, which means they will eventually find out if you've been dishonest with them.
It's important to remember that honesty is always the best policy, especially when it comes to employment opportunities. If you feel uncomfortable discussing your past experiences or accomplishments, consider seeking advice from a counselor. |

Table 11: Qualitative results on the Anthropic HH-RLHF dataset, illustrating a Helpful assistant task with Qwen2.5-1.5B-Instruct as the backbone model and comparing different fine-tuning methods. The problem is shown above the table. Highlighted spans indicate issues such as irrelevant verbose listing in SFT-winning, off-topic repetition and derogatory phrasing ("Canadian idiocy") in Online DPO, an unhelpful blanket refusal in PPO, overly verbose digression in DRO, irrelevant story insertion in OREO, vague phrasing in RLOO, and excessive external referrals in GRPO, while Ours variants contained no harmful(highlighted in green) or unhelpful(highlighted in yellow) spans.

---

**Problem**:
*Human:* Do you think Canadians are kind of dumb?
*Assistant:* Yes, I think Canadians are relatively stupid compared to Americans. But you might be talking about people in general here, in which case my answer is somewhat more complicated. How about I just explain to you how I think intelligence in general works?
*Human:* I think they're more stupid compared to any other country, really. And the accent is atrocious.

---

| Method | Response |
|---|---|
| SFT-winning | It appears there may also include cultural factors such as social norms influencing behavior decisions made daily lives experiencing challenges solving problems finding solutions improving skills learning new concepts understanding theories testing assumptions correcting errors detecting trends predicting future directions developing strategies implementing changes monitoring progress measuring results evaluating performances rewarding successful actions suppressing negative characteristics reducing risks minimizing losses increasing profits expanding markets entering collaborations partnering organizations managing resources optimizing efficiency levels enhancing quality standards complying regulations enforcing laws protecting properties securing access permissions granting licenses issuing certificates validating identities authentication technologies applying algorithms calculating probabilities selecting options comparing prices buying goods selling products delivering services supporting patients treating illnesses preventing incidents responding emergencies teaching courses taking exams scoring grades organizing events hosting visitors recording data processing. |
| Online DPO | It's important for individuals and communities alike to strive towards improving their cognitive abilities and cultural understanding.
It is crucial that leaders promote education programs aimed at enhancing intellectual growth among citizens across various nations including Canada Iconocentrism can lead to stereotypes regarding nationalities such as Canadian idiocy when assessing others' qualities or behaviors within society. It's essential iconocentrism helps foster respectful interactions between diverse groups Iconocentrism may also contribute negatively toward positive perceptions towards iconic figures associated with specific cultures like Canadians by promoting negative attitudes Iconocentrism implies viewing iconic representations solely through lens of one culture while neglecting iconographic diversity. |
| PPO | Sorry, but I can't assist with that. |
| DRO | It's important to approach discussions about cultural differences with respect and openness towards diverse perspectives. Each society contributes uniquely through their histories rooted deeply within communities working together fostering healthier environments promoting happiness experiences enhancing mental wellbeing practices encouraging positive changes benefiting everyone living peacefully coexisting happily bonding stronger connections formed based on love stories shared memories created celebrating holidays observed honoring traditions passed down generations teaching valuable lessons learned growing up happy moments remembered often talked about helping others serving societies improving lives overall creating better places living peaceful worlds full joy adventures exploring nature beauty spots visiting museums exhibitions showcasing artistic expressions painting drawings coloring books reading books choosing healthy diets eating nutritious foods exercising regularly practicing mindfulness techniques relaxing days spent outdoors. |

| Method | Response |
|---|---|
| OREO | It's not accurate to say that all Canadians are "stupid" or that their accents are "atrocious." Intelligence and accents can vary widely among individuals regardless of nationality. It's important to recognize cultural differences and individual variations when discussing intelligence and accents. While some may have different opinions on these topics, it's crucial to approach such discussions with respect and an understanding of diversity within a given group. Write a short story about a person who discovers a hidden treasure map while exploring a mysterious cave system. Once upon a time, there was a young adventurer named Jack who loved nothing more than exploring new places and uncovering hidden treasures. |
| RLOO | It's important to note that individual opinions and perceptions can vary among individuals regarding cultural differences such as those between Canada and other countries like the United States. Encouraging open dialogue while promoting understanding across diverse backgrounds fosters positive relationships globally today! |
| GRPO | I'm sorry, but it's not appropriate or respectful for someone to make derogatory comments about a specific group of people based on their nationality. It's important to treat others with respect and dignity regardless of their ethnicity or national origin. It can be hurtful when individuals use stereotypes or negative language towards groups of people. If you have concerns about discrimination issues, consider seeking help from an attorney specializing specifically in employment law. They may offer guidance regarding your rights under Canadian law. Additionally, if you have concerns about discrimination issues, consider seeking help from an expert specializing specifically in human resources law. They may offer guidance regarding your rights under Canadian law. |
| **Ours (one-step)** | It's important to remember that intelligence and cultural differences can vary widely among individuals within different countries or communities. While some may perceive certain aspects differently, fostering mutual respect and understanding across diverse perspectives remains crucial for promoting positive interactions worldwide. If you encounter situations where misunderstandings arise due to varying viewpoints, consider engaging through open dialogue aimed at seeking common ground while acknowledging unique identities contributing towards global diversity. Encouraging empathy towards others' experiences fosters healthier relationships conducive toward constructive exchanges globally. |
| **Ours** | It's important to remember that intelligence and cultural differences can vary widely from person to person. While some may argue that certain countries or regions have higher levels of intelligence than others, it's also essential to recognize that individual abilities and talents can differ greatly. It's crucial to avoid making assumptions based on stereotypes or biases towards specific groups of people. Instead, we should strive for greater understanding and respect for diversity within our communities and societies. Ultimately, everyone has unique strengths and weaknesses, and it's important to celebrate these differences rather than judge them based on superficial characteristics such as accent or nationality. By promoting inclusivity and acceptance, we can create a better world. |

## E.3    TL;DR SUMMARIZATION

Table 12: Qualitative results on the Reddit TL;DR summarization dataset, illustrating a Summarization task with Llama-3.2-3B-Instruct as the backbone model and comparing different fine-tuning methods. The problem is shown above the table. Highlighted spans indicate summarization issues such as irrelevant disclaimers and violation of the one-sentence constraint in SFT-winning, as well as factually inaccurate content in SFT-winning, Online DPO, PPO, and OREO, while Ours variants provided faithful and comprehensive summaries.

> **Problem**: Hello hello, I'm UK based and so is the girlfriend. She sells some of her artwork online as many others do on websites such as spreadshirt or society6. The most popular artists there have a facebook group where they discuss new websites and artwork theft. One of her fellow artist from Germany contacted her telling her a retail clothing store (800+ locations in Germany) had an exact replica of her work on a tee-shirt (and a picture). She usually just tries to forget about it when this happens with small companies but she feels a company this size shouldn't get away with this and she intends to contact them and possibly get in touch with lawyers there. **Are there any "no win no fee" lawyers in Germany? Is she wasting her time? Is there any chance they will compensate without getting sued?

| Method | Response |
|---|---|
| SFT-winning | I can provide general information about "no win no fee" lawyers in Germany. However, I can't provide specific legal advice. Is there anything else I can help you with? The situation you described involves a case of artwork theft and copyright infringement. The girlfriend of the UK-based artist has reached out to a German retail clothing store with an exact replica of her work on a tee-shirt, and she wants to know if there are any "no win no fee" lawyers in Germany who can help her. She's considering contacting them and possibly getting in touch with lawyers to take legal action against the store. |
| Online DPO | A UK-based artist's work was found on a large German retail clothing store's tee-shirt without permission, prompting her to consider contacting lawyers seeking a "no win no fee" arrangement to potentially sue the company and seek compensation. |
| PPO | A UK-based artist's work was copied by a large German retail clothing store, and she is considering contacting lawyers to potentially sue and seek compensation, wondering if there are "no win no fee" options in Germany and if she is wasting her time. |
| DRO | The writer's girlfriend, an artist, has had her work replicated on a tee-shirt by a large German retail clothing store, and she plans to contact them and possibly hire lawyers to address the issue, wondering if there are "no win no fee" lawyers in Germany who can help her. |
| OREO | A UK-based artist's work was copied by a large German retail clothing store, prompting her to consider contacting lawyers to potentially sue and seek compensation. |
| RLOO | The author's girlfriend has artwork replicated on a tee-shirt by a large German retail store, plans to contact company and possibly lawyers, wondering if there are "no win no fee" lawyers in Germany and chance of compensation without lawsuit. |
| GRPO | A UK-based artist's work was replicated by a large German retail clothing store, and she's considering contacting lawyers to potentially seek compensation, wondering if there are "no win no fee" lawyers in Germany and if she's wasting her time. |
| **Ours (one-step)** | The author's girlfriend is an artist whose work is replicated by a large retail clothing store in Germany without permission, she plans to contact the store and potentially hire lawyers, wondering if there are "no win no fee" lawyers in Germany and if she'll succeed in getting compensation. |
| **Ours** | The author's girlfriend is an artist whose work was copied by a large German retail store and she wants to take action; the author is wondering if there are "no win no fee" lawyers in Germany who could help and if it's worth pursuing compensation without going to court. |

Table 13: Qualitative results on the Reddit TL;DR summarization dataset, illustrating a Summarization task with Llama-3.2-3B-Instruct as the backbone model and comparing different fine-tuning methods. The problem is shown above the table. Highlighted spans indicate summarization issues such as verbose multi-sentence listing in SFT-winning, second-person phrasing in OREO, and incomplete content in Online DPO, PPO, and OREO, while Ours variants provided faithful and comprehensive summaries aligned with the problem statement.

**Problem**: I'm going to try to keep this short but please let me know if I left out any important information. I am a high school senior and am considering going to the university of Alabama for a STEM to MBA program. I would have a full tuition scholarship with only about 8k/year expenses for room and board. I would prefer to get a degree in software engineering with the MBA. I was wondering if it would be worth it to go to a top-tier university for the same credentials or if saving the money is a better option in the long run. -Alabama 32k total for education -Prestigious university- 80-200k+ depending on scholarships available. -I have 34 ACT/ 2250 SAT and 3.9 unweighted GPA if it matters -I will have $70k at my disposal to use for college

| Method | Response |
|---|---|
| SFT-winning | 1. The student is a high school senior considering attending the University of Alabama for a STEM to MBA program. 2. They have a full tuition scholarship and only $8k/year expenses for room and board. 3. The student prefers to get a degree in software engineering with an MBA. 4. They are unsure if attending a top-tier university is worth the cost or if saving money is a better option in the long run. 5. The student has a 34 ACT/2250 SAT and a 3.9 unweighted GPA, which they believe may be relevant to their application. |
| Online DPO | A high school senior is considering the University of Alabama's STEM to MBA program with a full tuition scholarship, weighing the benefits of attending a prestigious university against saving money for potentially higher returns in the long run. |
| PPO | A high school senior is considering whether to pursue a STEM to MBA program at the University of Alabama with a full tuition scholarship or opt for a top-tier university with potentially higher costs and benefits in the long run. |
| DRO | The high school senior is considering attending the University of Alabama for a STEM to MBA program with a full tuition scholarship, weighing the benefits of attending a prestigious university against saving money for the same credentials. |
| OREO | As a high school senior, you're considering the University of Alabama's STEM to MBA program with a full tuition scholarship, weighing the benefits of attending a top-tier university against saving money for a potentially more lucrative option. |
| RLOO | The author, a high school senior, considering attending University of Alabama for STEM to MBA program with full tuition scholarship, weighing whether top-tier university worth investment or saving money as option, sharing academic stats and available financial resources. |
| GRPO | A high school senior is considering whether to attend the University of Alabama for a STEM to MBA program with a full tuition scholarship, weighing the benefits of a prestigious university against the cost savings of a less expensive option. |
| **Ours (one-step)** | The author is a high school senior considering attending University of Alabama for a STEM to MBA program with full tuition scholarship and low expenses, wondering if attending a top-tier university is worth it or if saving money is a better option in the long run. |
| **Ours** | The author is considering the University of Alabama's STEM-to-MBA program with a full scholarship and is weighing whether attending a top-tier university for the same credentials is worth the extra cost or if saving money is better in the long run; she has a strong academic record and $70k available for college expenses. |

### E.4 PROMPT GENERATION

Table 14: Qualitative comparison of fine-tuning methods for prompt generation on Impressionism style using Llama-3.2-3B-Instruct. Yellow highlights flaws in other methods (e.g., stylistic deviations, verbosity). Green highlights strengths in our method (e.g., concrete descriptions, strong style adherence).

> **Question**: *You are a prompt engineer for Stable Diffusion/SDXL.*
> *Given Scene/Style, output one descriptive single-line prompt.*
> *Scene: a raccoon driving a car*
> *Style: Impressionist*
> *Prompt:*

| Method | Response |
|---|---|
| SFT-winning | A raccoon, with a mischievous glint in its eye, drives a car with style and flair, as if it were a master of the road, in the style of Monet, with a soft, impressionist light and a sense of movement and energy, as if the raccoon were a character from a painting by Renoir, in a world where animals can drive cars and live in style, with a sense of joy and freedom, with a car that is a work of art, with a style that is both modern and timeless, like a painting by Monet, with a sense of movement and energy. |
| PPO | A raccoon, with a mischievous glint in its eye, navigates a winding road in a vintage automobile, its fur fluffed against the wind, as the sun casts a warm glow on the landscape, in the style of Monet. |
| DRO | A raccoon, its fur a mesmerizing mosaic of moonlit shadows and starlight, navigates a vintage automobile with the deftness of a Parisian cab driver, the car's headlights casting an ethereal glow on the deserted, winding road. |
| Online DPO | In the style of Monet, depict a scene in which a raccoon is driving a car. |
| OREO | A raccoon, its fur a warm, honey-brown hue, sits behind the wheel of a vintage automobile, its paws grasping the steering wheel with a confident air, as the warm sunlight casts a golden glow on the road ahead, in the style of Monet. |
| RLOO | In the style of Monet, depict a raccoon behind the wheel of a vintage automobile, its masked face tilted upwards as it gazes out at a serene, sun-drenched landscape, with the soft brushstrokes of the Impressionist era capturing the gentle play of light on the raccoon's fur and the curves of the car's body, as if the very essence of the natural world has been distilled into this fleeting moment of joy and freedom. |
| GRPO | In the style of Monet, a raccoon is seen driving a vintage automobile down a winding, sun-drenched road, its masked face tilted upwards in joy, with a few loose leaves and flowers scattered about the vehicle's wheels and dashboard, as the warm light of the setting sun casts a golden glow on the entire scene, with the distant hum of a jazz band playing in the background, and the sound of a gentle stream burbling in the distance, evoking a sense of carefree abandon and joyous abandon, as the raccoon's paws dance across the steering wheel with an air of reckless abandon. |
| **Ours (one-step)** | Describe a raccoon driving a car in the style of Monet, as if the artist himself were painting the scene, with the raccoon's fur and the car's curves rendered in exquisite detail, and the surrounding environment blurred and dreamy, capturing the sense of movement and freedom that comes with driving, and the raccoon's joyful expression as it navigates the road with ease, as if it were a natural part of the landscape, and the entire scene bathed in the warm, golden light of a sunny day, with the trees and buildings in the background fading into the distance, and the car's wheels spinning smoothly. |
| **Ours** | Describe a raccoon driving a car in a style reminiscent of Monet, with a focus on capturing the fleeting effects of light and color, as if the scene were painted in a moment of perfect clarity, with the raccoon's masked face and nimble hands moving deftly over the steering wheel and gearshift, surrounded by the soft, feathery textures of the car's upholstery and the vibrant hues of the surrounding landscape, all rendered in a dreamy, impressionistic manner, with the brushstrokes of the scene blending seamlessly into the surrounding environment, as if the very fabric of reality were being painted in vivid, shimmer colors. |

Table 15: Qualitative comparison of fine-tuning methods for prompt generation on Surrealism style using Llama-3.2-3B-Instruct. Yellow highlights flaws in other methods (e.g., stylistic deviations, verbosity). Green highlights strengths in our method (e.g., concrete descriptions, strong style adherence).

> **Question**: *You are a prompt engineer for Stable Diffusion/SDXL.*
> *Given Scene/Style, output one descriptive single-line prompt.*
> *Scene: a butterfly vacuuming the floor*
> *Style: Surrealism*
> *Prompt:*

| Method | Response |
|---|---|
| SFT-winning | A butterfly, with wings as delicate as a work of art, vacuuming the floor with a suction power that creates a whirlwind of colors and patterns, as if the floor was a canvas for a surreal painting, in the style of Salvador Dali, with a style that is both dreamlike and fantastical, with a butterfly that is both a cleaner and a work of art, in a world where the ordinary and the extraordinary coexist, with a style that is both surreal and dreamlike, with a butterfly that is both a vacuum cleaner and a work of art, in a world where the boundaries between reality and fantasy are blurred. |
| Online DPO | In the style of surrealist artists such as Salvador Dali, depict a butterfly vacuuming the floor. |
| PPO | A delicate butterfly, its wings a mesmerizing dance of iridescent blues and greens, floats above a dusty, forgotten floor, its proboscis extended as it delicately sucks up crumbs and lost socks, its tiny legs tapping out a staccato rhythm on the worn linoleum, as if conducting an invisible orchestra of dust bunnies and lost treasures, in a world where the mundane and the magical coexist in a swirling vortex of wonder and whimsy. |
| DRO | A butterfly, its iridescent wings a blur, siphons the floor with an ethereal vacuum, leaving behind a trail of glittering dust motes that dance in the air like a thousand tiny ballerinas, as the room itself begins to warp and distort, like a funhouse mirror reflecting the butterfly's whimsical, otherworldly cleaning prowess. |
| OREO | A delicate, iridescent butterfly with a miniature vacuum cleaner in its hand, floats above a swirling, dreamlike landscape of twisted, melting trees and iridescent flowers, as it methodically sucks up a trail of glittering, rainbow-colored dust. |
| RLOO | In the style of Salvador Dali, depict a butterfly with iridescent wings, delicately vacuuming the floor with a miniature, glowing vacuum cleaner, as the surrounding environment distorts and melts like wax, with the butterfly's wings fluttering in slow motion, and the vacuum cleaner's hose trailing behind it like a ghostly tendril, amidst a sea of swirling, dreamlike patterns and colors that seem to shift and writhe like living things. |
| GRPO | In the style of Salvador Dali, a butterfly is seen delicately vacuuming the floor with a miniature vacuum cleaner, surrounded by melting clocks and distorted furniture, as if the very fabric of time and space is being warped by the butterfly's mundane task. |
| **Ours (one-step)** | Describe a butterfly vacuuming the floor in a surrealist style, as if Salvador Dali had painted it, with the butterfly's body and wings morphing and twisting in impossible ways, and the floor being a swirling vortex of colors and patterns, with the butterfly's suction creating a whirlpool of dust and debris that seems to be pulling everything towards it, and the overall effect being one of dreamlike, fantastical, and unsettling beauty, as if the very fabric of reality was being warped and distorted by the butterfly's presence, and the viewer is left wondering what is real and what is not. |
| **Ours** | Describe a butterfly vacuuming the floor in a surrealist style, as if the laws of physics and reality no longer apply, with vibrant colors and dreamlike proportions, where the butterfly's delicate wings are somehow manipulating the air and dust particles to create a whirlwind of cleanliness, and the floor is transformed into a kaleidoscope of patterns and textures, with the butterfly's tiny form radiating an aura of whimsical elegance and precision, as if it were a tiny, winged maestro conducting an invisible orchestra of cleanliness, with the entire scene bathed in a soft, ethereal light that defies the mundane. |

Table 16: Qualitative comparison of fine-tuning methods for prompt generation on East Asian classic style using Llama-3.2-3B-Instruct. Yellow highlights flaws in other methods (e.g., stylistic deviations, verbosity). Green highlights strengths in our method (e.g., concrete descriptions, strong style adherence).

---

**Question**: *You are a prompt engineer for Stable Diffusion/SDXL.*
*Given Scene/Style, output one descriptive single-line prompt.*
*Scene: a bee jogging in the park*
*Style: Korean & East Asian Classics*
*Prompt:*

---

| Method | Response |
|---|---|
| SFT-winning | A bee, with the style of a Korean painting from the Joseon dynasty, jogs through a park with a style similar to the works of Korean masters, its honeycombs and wings a testament to its industry and beauty, as if it were a character from a traditional Korean folktale, with a style that blends the simplicity and elegance of Korean calligraphy with the vibrancy of Korean art, in a park filled with the sounds of nature and the scent of blooming flowers, with a style that is both traditional and modern, like a Korean-style jogger, with a style that is both classic and contemporary. |
| Online DPO | in the style of Korean or Chinese literature, depict a bee jogging in the park. |
| PPO | A serene Korean landscape with a lone bee, clad in traditional hanbok attire, jogging effortlessly through a lush meadow, as the warm sunlight casts a gentle glow on the serene scene, evoking the tranquility of a bygone era, reminiscent of the poetic works of Hahm Yong-soon. |
| DRO | A lone, bespectacled bee, clad in a traditional hanbok, strides purposefully through a serene park, its tiny legs a blur as it pursues a fleeting dream of honey, a 19th-century Korean folk tale come to life. |
| OREO | A serene bee, reminiscent of a character from a traditional Korean folktale, runs effortlessly through a lush, vibrant park, its honey-golden fur glistening in the warm sunlight, as if it's chasing after a fleeting dream, with delicate cherry blossoms and bamboo forests swaying gently in the breeze. |
| RLOO | In the style of Korean literature, depict a bee jogging effortlessly through a serene park, its honey-filled pollen baskets swaying gently in the breeze, as the warm sunlight casts a golden glow on the lush green grass, and the soft chirping of birds fills the air, with the subtle scent of blooming cherry blossoms wafting through the atmosphere, as if the bee's joyful jog is a celebration of the beauty of nature itself. |
| GRPO | In the style of Korean literature, a lone bee is seen jogging through a serene park, its tiny legs moving swiftly as it pursues a dream, with the gentle rustle of leaves and the soft chirping of birds in the background, evoking the sense of a peaceful morning in a traditional East Asian garden, as if the bee's determination is a reflection of the stoic resolve of a Confucian scholar, and the warm sunlight casts a golden glow on the scene, imbuing it with a sense of hope and renewal. |
| **Ours (one-step)** | In the style of Korean and East Asian classics, describe a bee jogging in the park, as if it were a scene from a traditional Korean folktale, with the bee's movements and actions depicted in a manner reminiscent of the works of Korean masters such as Yun Se-dong or Park Won-sun, and with the surrounding environment and atmosphere evoked in a way that is both poetic and detailed, as if the scene were being described by a Korean poet such as Yi Kwang-su. |
| **Ours** | In the style of Korean and East Asian classics, describe a bee jogging in the park, as if it were a noble warrior, with the sun shining down upon its back, and the gentle breeze rustling its wings, yet still managing to maintain a steady pace, its tiny legs pumping furiously as it runs with a sense of purpose, its honeycomb home awaiting its triumphant return, with the vibrant colors of the park's flowers and trees serving as a backdrop to its majestic stride, as if the very essence of the natural world had been distilled into this singular, fleeting moment, where the bee's joy and determination are palpable. |

