# OpenReview forum: "Group-Normalized Implicit Value Optimization for Language Models"
_ICLR.cc/2026/Conference — ICLR 2026 Poster_

### Official Review · Reviewer_7Q1n · 2025-10-23

**Soundness:** 3
**Presentation:** 3
**Contribution:** 3
**Rating:** 6
**Confidence:** 3

**Summary:**

This paper introduces Group-Normalized Implicit Value Optimization (GN-IVO), a critic-free policy-gradients loss to optimize the KL-regularized RL objective common in LLM post-training at the token level. They achieve this by first defining a value function as an expectation of the reward over possible continuations of a sequence, and 1) using a list-ranking loss to approximate the relative ordering of the values in a group of responses to get rid of the intractable partition function, and 2) using the DPO trick of using the new and old policy log-likelihood ratios as an estimator of the value function.  The main advantage of the technique is that it provides an explicit-credit assignment distribution of the reward over specific tokens. The approach is validated in mathematical reason and text generation benchmarks.

**Strengths:**

- S1) The paper tackles an important problem when seeking to improve sampling efficiency by explicitly computing the credit-assignment distribution over the individual actions.
- S2) The method is solidly motivated
- S3) The method is evaluated on 2 different kinds of tasks showing consistent improvements.

**Weaknesses:**

- W1) While the paper considers a wide range of baselines, I find that the unbiased version of GRPO --Dr. GRPO-- is prominently missing.
- W2) I'm not convinced on the performance metrics of section 4.2. The authors use avg. reward as a metric in different LLM post-training task, but this is not necessarily the same as good performance. For example, once can have a collapsed model that outputs high-reward samples according to the reward function, but where the LM could have degenerated in various ways. The alpha parameter is controlling for this trade-off which is particularly important for the experiments used in Section 4.2. Better metrics could be an LLM judge and/or KL divergence to the target goal probability (Eq. 2).

I'm happy to revise my score upwards if these concerns and other concerns described in the questions below are addressed.

**Questions:**

- Q1) I'm not sure I understand the need to start from Eq. 7. Couldn't have you derived your objective from $CE(V, V_\psi)$ where the cross-entropy is computed across a batch of K samples?
- Q2) Theorem 3.2: "equivalently...", I think there is a typo in $yt$, and should be $y<t$, right?
- Q3) In the loss of GN-IVO, you write that t is sampled from $U\{1...T\}$. Why not directly a sum? And how many t's you sample in practice?
- Q4) What is the effect of not normalizing $\exp(R/\alpha)$? I wonder if the normalization induces some bias in the algorithm?
- Q5) Sorry if I missed this, but what value of $\alpha$ are you using in your experiments?
- Q6) GRPO has been observed to generate less-diverse models (e.g., https://arxiv.org/abs/2506.02355v1). Is it the case for models trained using your proposed loss? To see this, you could report, say, Pass@128. Less-diverse models struggle at leveraging the additional sampling budget.
- Q7) I'm surprised by what you call "rejection sampling". For me, rejection sampling would be generating a bunch of candidates, retaining only the high reward ones, and do SFT on those (e.g. https://arxiv.org/abs/2203.14465)
- Q8) L419: What does a single-step objective mean in your case? You say $T=1$, but that reads as if there would be just one token to me.
- Q9) I'm not sure I agree with the discussion on the temperature coefficient in section 4.3. "We observe that the model trains effectively with lower temperature values (α = 0.1, 0.2). In contrast, for higher values such as α = 0.5 and 1.0, learning is significantly slower, and the models ultimately achieve lower performance. This is because a more uniform target distribution provides a weaker and less discriminative training signal, making it difficult for the policy to distinguish superior from mediocre responses." => As noted in Eq. 2, different values of alpha correspond to different target distributions and so, higher values of alpha correspond to lower expected reward. Couldn't the effect of $\alpha$ be explained by different target distributions instead of different learning dynamics?
- Q9) Proof of Theorem 3.1: why the integrals? These are discrete values...
- Q10) How does this distribution matching approach differ from others such as https://arxiv.org/abs/2302.08215, https://arxiv.org/abs/2012.11635 or https://arxiv.org/abs/2205.14219 ?
- Q11) I think your internal value-ranking loss is equivalent to ListNet: from "Learning to Rank: From Pairwise Approach to Listwise Approach". (https://www.microsoft.com/en-us/research/wp-content/uploads/2016/02/tr-2007-40.pdf) Do you agree?

---

> ### Author Response · Authors · 2025-11-21
>
> **(Weakness 1) Comparison with Dr.GRPO**
>
> Dr.GRPO is a variant of GRPO that computes the advantage without standard deviation normalization.
> We conducted additional experiments to evaluate this, and the results are as follows.
>
> | Model/Method | AMC23 P@1 | P@3 | MM P@1 | P@3 |OB P@1 | P@3 | AIME24 P@1 | P@3 | AIME25 P@1 | P@3 |
> |:---|---:|---:|---:|---:|---:|---:|---:|---:|---:|---:|
> | Llama-3.1-8B-Instruct |  |  |  |  |  |  |  |  |  |  |
> | Dr.GRPO | 27.5 | 45.0 | 24.2 | 35.3 | 16.0 | 27.1 | 13.3 | 16.6 | 0.0 | 6.6 |
> | ours         | 42.5 | 45.0 | 26.1 | 36.0 | 17.3 | 27.8 | 10.0 | 16.6 | 3.3 | 3.3 |
> | Qwen2.5-Math-7B |  |  |  |  |  |  |  |  |  |  |
> | Dr.GRPO | 52.5 | 72.5 | 27.5 | 40.8 | 33.3 | 47.7 | 16.6 | 26.6 | 10.0 | 13.3 |
> | ours         | 62.5 | 75.0 | 31.6 | 41.9 | 39.8 | 49.0 | 30.0 | 40.0 | 13.3 | 23.3 |
>
> Compared to Dr.GRPO, our algorithm consistently achieves better performance than Dr.GRPO.
>
> **(Weakness 2) LLM-based evaluation in section 4.2**
>
> Following your suggestion, we performed an additional LLM-based evaluation.
> We utilized GPT-4.1 as a judge to compare responses generated by our method against those from the baselines and determine whether our method’s response was a win/tie/loss.
> The corresponding results are reported as percentages in the table below.
>
> | Model/Method       | HH    |      |      | TL;DR |      |      | Prompt |      |      |
> |:---------------------|:-----:|:----:|:----:|:-----:|:----:|:----:|:------:|:----:|:----:|
> |                      | win   | tie  | loss | win   | tie  | loss | win    | tie  | loss |
> | Llama-3.2-3B-Instruct     |       |      |      |       |      |      |        |      |      |
> | vs SFT-w             | 82.5  | 6.8  | 10.5 | 98.7  | 0.3  | 0.9  | 70.1   | 0.0  | 29.8 |
> | vs Online DPO        | 93.2  | 2.5  | 4.1  | 44.8  | 12.6 | 42.4 | 99.8   | 0.0  | 0.1  |
> | vs PPO               | 75.3  | 11.6 | 12.9 | 48.7  | 8.1  | 43.1 | 17.7   | 0.0  | 82.2 |
> | vs DRO               | 42.4  | 33.6 | 23.8 | 34.1  | 23.2 | 42.6 | 53.6   | 0.0  | 46.3 |
> | vs OREO              | 79.9  | 8.0  | 12.0 | 60.7  | 6.4  | 32.8 | 17.1   | 0.0  | 82.8 |
> | vs RLOO              | 84.6  | 6.0  | 9.2  | 43.4  | 25.8 | 30.7 | 72.9   | 0.0  | 27.0 |
> | vs GRPO              | 73.7  | 18.1 | 8.1  | 47.2  | 10.8 | 41.8 | 58.3   | 0.0  | 41.6 |
> | Qwen2.5-1.5B-Instruct  |       |      |      |       |      |      |        |      |      |
> | vs SFT-w             | 99.8  | 0.0  | 0.1  | 99.8  | 0.0  | 0.1  | 100.0  | 0.0  | 0.0  |
> | vs Online DPO        | 64.9  | 20.6 | 14.4 | 75.6  | 0.1  | 24.1 | 45.1   | 20.3 | 34.4 |
> | vs PPO               | 90.5  | 0.3  | 9.1  | 98.8  | 0.0  | 1.1  | 52.5   | 0.3  | 47.1 |
> | vs DRO               | 58.5  | 30.4 | 10.9 | 50.3  | 3.8  | 45.8 | 17.1   | 16.8 | 66.0 |
> | vs OREO              | 75.8  | 0.6  | 23.5 | 51.6  | 4.9  | 43.4 | 51.4   | 0.0  | 48.5 |
> | vs RLOO              | 41.5  | 25.9 | 31.5 | 50.8  | 3.8  | 45.3 | 52.7   | 0.4  | 46.7 |
> | vs GRPO              | 56.3  | 1.6  | 33.0 | 55.8  | 10.0 | 34.1 | 59.3   | 20.0 | 20.6 |
>
> **(Q1)**
> We start from Eq.(7) to leverage the linearity of expectation.
> In CE(V, V_\psi), the linearity of expectation cannot be applied to the term in the denominator.
> To be specific:
> $$\frac{ e^{ V( x, y^{(i)}\_{<t} ) }}{  \sum\_j e^{ V( x, y^{(j)}\_{<t} ) } }
> = \frac{ E\_{\pi\_{ \theta\_\text{old}} (y|y\_{<t}) } [e^{ R( x, y^{(i)} ) } ] }{  \sum\_j E_{ \pi\_{\theta\_\text{old}} (y|y\_{<t}) } [ e^{ R( x, y^{(j)} ) } ]  } \neq E\_{\pi\_{ \theta\_\text{old}} (y|y\_{<t}) }  [ \frac{ e^{ R( x, y^{(i)} ) }}{  \sum\_j e^{ R( x, y^{(j)} ) } } ].
> (E[X]/E[Y] \neq E[X/Y])
> $$
>
> If we have misunderstood your question, please let us know.
>
> **(Q2)** Thank you for your point. $y\_t$ should be $y\_{<t}$.
>
> **(Q3)** We sample $t$ 5 times for math reasoning and 20 times for text generation tasks.
> Based on your question and comments from other reviewers, we further investigated the effect of averaging over all time-steps in the HH-RLHF.
> Specifically, we conducted experiments by varying the number of sampled time steps across {1, 20, $T$} (where $T$ represents the full sequence length). The results are shown in the table below.
>
> | # sampled steps \(t\) | 1    | 20   | T (full) |
> |:--------------------------:|:----:|:----:|:-------------:|
> | Ours ( score ↑)          | 1.40 | 1.65 | 1.80         |
>
> The training curve is shown in Figure 7. (Appendix, page 19)
>
> **(Q4)** This is illustrated in Figure 6 (Appendix, page 18).
>
> **(Q5)** We use 0.2 for alpha. (Table 5, Appendix, page 19)

---

> ### Author Response · Authors · 2025-11-21
>
> **(Q6)** We report the Pass@32 and Pass@128 metrics in the table below.
> | Model/Method          | AMC23      |        | MM         |        | OB         |        | AIME24     |        | AIME25     |        |
> |:----------------|:----------:|:------:|:----------:|:------:|:----------:|:------:|:----------:|:------:|:----------:|:------:|
> |  Qwen2.5-Math-7B  | P@32       | P@128  | P@32       | P@128  | P@32       | P@128  | P@32       | P@128  | P@32       | P@128  |
> | GRPO            | 90.0       | 90.8 (+0.8)  | 60.2   | 63.7 (+3.5) | 69.3   | 76.4 (+7.1)  | 50.0   | 60.0 (+10.0) | 40.0   | 40.0 (+0.0)  |
> | ours            | 95.0       | 95.0 (+0.0)  | 58.4   | 66.5 (+8.1)   | 69.4   | 77.6 (+8.2)  | 56.6   | 76.6 (+20.0) | 33.3   | 43.3 (+10.0) |
>
> Generally, our method demonstrates larger improvements from P@32 to P@128 compared to GRPO.
>
> **(Q7)** We rename Rejection Sampling to SFT-winning.
>
> **(Q8)** L419 (T=1)
>
> A one-step variant is the bandit setting, which treats the full generated sequence as a single-step action. Consequently, the objective does not involve an expectation over timestep ($t$) and only considers the full sequence.
>
> **(Q9)** The effect of $\alpha$
>
> We agree. The effect of $\alpha$ is explained by the shape of target distributions.
> Higher values (e.g., $\alpha=0.5, 1.0$) yield a more uniform target distribution.Consequently, the policy assigns comparable probability mass to both superior and mediocre response, which leads to the lower expected rewards.
>
> **(Q10)** We replace the integral with a summation.
>
>
> **(Q11-1) Comparison with [1]**
>
> DPG [1] is a policy-matching objective that aims to minimize the KL divergence between the target $\pi\_{\theta^\star}(\cdot)\propto \pi\_\theta(\cdot) e^{R(\cdot)/\alpha}$ and the learned policy $\pi\_\theta$,
> i.e.,
>
> $
> \nabla\_\theta D\_\text{KL} (\pi\_{\theta^\star} | |\pi\_\theta) =  \nabla\_\theta CE (\pi\_{\theta^\star} || \pi\_\theta)  = - \mathbb{E} \_{y \sim \pi\_\theta(y|x)} [\frac{\pi\_{\theta^\star} (y|x) }{\pi\_\theta(y|x) } \nabla\_\theta \log \pi\_\theta(y|x) ].
> $
>
> Since sampling directly from the target $\pi\_{\theta^\star}$ is not possible, DPG relies on Importance Sampling (IS).
> If self-normalized IS is used, the IS term $\frac{\pi\_{\theta^\star}(y|x)}{\pi\_\theta(y|x)}=e^{R(x,y)/\alpha}/Z(x)$ becomes the softmax reward. However, there are fundamental differences.
> We do not attempt to match the learned policy to the target policy $\pi\_{\theta^\star}$ over $y$.
> Instead, we implicitly fit the value $V$ by matching our policy’s induced distribution to an empirical target distribution defined over a group, given by the softmax of their values $\exp(V^i)/\sum_j \exp(V^j)$.
> This distinction leads to a different loss. The DPG uses the standard policy gradient term $\nabla\_\theta \log \pi\_\theta(y|x)$. Our objective uses a group-normalized policy term (the log-softmax over  $K$, $\nabla_\theta (\log \frac{ \pi\_{\theta}(y^{(i)}|x) }{ \pi\_{\theta\_\text{old}}(y^{(i)}|x)   } -\log \sum_j \frac{\pi\_{\theta} (y^{(j)}|x)  }{\pi\_{\theta\_\text{old}} (y^{(j)}|x) } )$.
>
> [1] A Distributional Approach to Controlled Text Generation

---

> ### Author Response · Authors · 2025-11-21
>
> **(Q11-2) Comparison with [2]**
>
> [2] generalizes DPG [1] to a broader class of f-divergences. While our work focuses on the forward KL divergence between group-normalized distributions, the theoretical guarantees established in our work (Theorem 3.2, Consistency up to constant shift) extend to other f-divergences as well.
>
> An f-divergence $D\_f(p||q)$ is defined via a convex function $f$ such that $f(1) = 0$.
> Applying Jensen’s inequality, the divergence is non-negative for any distribution $p$ and $q$.
>
> $D\_f(p||q) = \sum_i q_i f(\frac{p_i}{q_i})\geq f(\sum\_i q\_i \frac{p\_i}{q\_i}) = f(\sum\_i p\_i) = f(1) = 0.$
>
> Equality holds if and only if $p\_i = q\_i$ for all $i$.
> In our formulation, the target distribution $p$ and the learned distribution $q\_\psi$ are defined as follows: (Here, we consider only $y$.)
>
> $ p\_i=\frac{ e^{R( x, y^{(i)}) } }{ \sum\_j e^{ R( x, y^{(j)} ) } },
> q\_{\psi,i}=\frac{ e^{R\_\psi( x,y^{(i)}) } }{\sum\_j e^{ R\_\psi ( x,y^{(j)} ) } }.$
>
> The equality condition $p\_i=q\_{\psi,i}$ implies that the learned values recover the soft value up to an additive constant, following the same logic as the proof in the appendix.
>
> We provide some f-divergence and our corresponding objective as below.
>
> | Name | $D_f (p \Vert q_\psi)$  | $\mathcal{L}_{f, \text{GN-IVO}} $ |
> |:---|:---|:---|
> | Forward KL | $ -H(p) - \sum_i p_i \log q_{\psi, i}  $ |  $-\sum_i \frac{ e^{R( x, y^{(i)}) } }{ \sum_j e^{ R( x, y^{(j)} ) } } (\log \frac{ \pi_{\theta}(y^{(i)}\mid x) }{ \pi_{\theta_\text{old}}(y^{(i)}\mid x)   } - \log{  \sum_j \frac{\pi_{\theta} (y^{(j)}\mid x)  }{\pi_{\theta_\text{old}} (y^{(j)}\mid x) } } )$ |
> | Reverse KL | $ -H(q\_{\psi}) - \sum_i q\_{\psi, i} \log p\_i $ |   $  - H(\cdot) - \sum_i \frac{  \frac{ \pi_{\theta}(y^{(i)}\mid x) }{ \pi_{\theta_\text{old}}(y^{(i)}\mid x)   } }{  \sum_j \frac{\pi_{\theta} (y^{(j)}\mid x)  }{\pi_{\theta_\text{old}} (y^{(j)}\mid x) } } \log  \frac{ e^{R( x, y^{(i)}) } }{ \sum_j e^{ R( x, y^{(j)} ) } }$
> | Total Variation | $ 1/2 \sum\_i \mid p\_i - q\_{\psi, i} \mid  $ | $  1/2 \sum_i  \mid \frac{  \frac{ \pi_{\theta}(y^{(i)}\mid x) }{ \pi_{\theta_\text{old}}(y^{(i)}\mid x)   } }{  \sum_j \frac{\pi_{\theta} (y^{(j)}\mid x)  }{\pi_{\theta_\text{old}} (y^{(j)}\mid x) } } -  \frac{ e^{R( x, y^{(i)}) } }{ \sum_j e^{ R( x, y^{(j)} ) } } \mid $
>
> **(Q11-3) Comparison with [3]**
>
> [3] aims to steer a fixed, pre-trained base model at inference time using token-level guidance without fine-tuning.
> Both methods derive a step-level relationship from a sequence-level one.
> While our work is formulated over the partial sequences,
> $\pi\_\theta(y\_{<t}|x)$, [3] focuses on the token conditional on a prefix, $\pi\_\theta(y_t|x,y_{<t})$.
>
> From our findings that $\pi\_{\theta^\star} (y\_{<t}|x) = \pi\_{\theta_\text{old}} (y\_{<t}|x) e^{V(x,y\_{<t})} / Z(x)$ , we can derive the corresponding relationship:
>
> $ \frac{\pi\_{\theta^\star} (y\_{<t+1}| x) }{ \pi\_{\theta^\star} (y\_{<t}|x) } ( = \frac{ \pi\_{\theta^\star} (y\_{t}, y\_{<t} | x) }{ \pi\_{\theta^\star} ( y\_{<t}| x ) } )
> = \frac{\pi\_{ \theta_\text{old} } (y\_{<t+1}|x) } { \pi\_{\theta\_\text{old}} (y\_{<t}|x) } \frac{ e^{ V(x,y\_{<t+1})} }{ e^{V(x,y\_{<t})} }.$
>
> $
> \rightarrow \pi\_{\theta^\star} (y\_{t}|x, y\_{<t}) = \pi\_{ \theta\_\text{old} } (y\_{t}|x, y\_{<t})
> \frac{ e^{V(x,y\_{<t+1}) } }{ e^{ V( x,y\_{<t} ) } }
> $
>
> This result shares a similar form to the one in [3].
>
> [3] trains an additional network $C$, (analogous to $V$) to guide the base model. Since [3] considers a binary constraint function, $C(x, y) \in \{0,1\}$, it learns $P(C(x,y)=1|x, y\_{<t})$ using a binary classifier.
> Because our work is designed for general, continuous rewards, the value function $V$ can be learned via a simple MSE loss within the framework of [3].
>
> **(Q12) Equivalence with ListNet**
>
> ListNet trains a probabilistic model to rank items via cross-entropy minimization.
> We agree that our group-normalized value loss is equivalent to the loss function used in ListNet.
> However, we emphasize that our work provides additional theoretical insights not covered in ListNet. Specifically, Theorem 3.2 and Corollary 3.3 characterize the theoretical properties of the values learned by this cross-entropy loss.
>
>
> We have revised our paper to clarify these points, and we appreciate your feedback.
>
> [2] Aligning Language Models with Preferences through f-divergence Minimization
>
> [3] Controllable Text Generation with Neurally-Decomposed Oracle

---

> > ### Comment · Reviewer_7Q1n · 2025-11-25
> >
> > Thank you for the very detailed answer!
> >
> > I have couple of clarification/follow up questions:
> >
> > 1. You say "Dr.GRPO is a variant of GRPO that computes the advantage without standard deviation normalization". It also removes the length normalization and sets the normalization constant $\beta=0$. I understand if you don't do the latter, as you are comparing different methods with equal KL regularization strength, but did you also take into account the (lack of) length normalization in your experiments?
> > 2. Incidentally, is your method undefined for $\alpha=0$ or is there some kind of limit?
> > 3. Looking at Table 5 in the appendix I see there are both $\alpha$ and $\beta$ parameters. I thought you were using $\alpha$ as a synonym of $\beta$, but if both parameters are used simultaneously could you clarify in which ways they differ? I'm sorry if I missed something obvious.

---

> ### Author Response · Authors · 2025-11-26
>
> **(Q1) Length normalization**
>
> We used the GRPOTrainer from ``trl library``, configuring the ``loss_type`` to ``dr_grpo``.
> We confirmed that this setting does not divide by the sequence as length N.
>
> Furthermore, our objective does not suffer from the length bias mentioned in Dr.GRPO, where the model reduces loss by generating long answers when wrong and short answers when right.
> This is avoided because our objective does not use the sequence length $N$.
>
> Simultaneously, we emphasize that our objective is based on relative value matching. In this framework $\pi_{\theta_\text{old}}$ acts as a mechanism for length normalization. Without $\pi_{\theta_\text{old}}$, longer sequences naturally yield lower probabilities $\prod_t \pi(y\_t| x, y\_{<t})$ .
> If $y^{(1)}$ happens to be longer than an incorrect $y^{(2)}$ (where $r^{(1}) > r^{(2)}$), the model would be forced to artificially boost the token probabilities of $y^{(1)}$ to overcome this length penalty. This can lead to model degeneration [4]. However, the inclusion of $\pi\_{\theta_\text{old}}$ in our objective avoids this.
> If we misunderstood your question, please let us know.
>
> [4] SimPO: Simple Preference Optimization with a Reference-Free Reward
>
>
> **(Q2) $\alpha \rightarrow 0$**
>
> We appreciate this insightful point.
> We have thought about the scenario where $\alpha \rightarrow 0$.
> In this limit, the target distribution converges to a one-hot distribution, concentrating the probability mass solely on the response with the largest reward. Consequently, the objective simplifies to:
>
> $\log\frac{ \pi\_\theta (y\_{<t}^{(w)} | x) } {\pi\_{\theta\_\text{old}} (y\_{<t}^{(w)}|x)} - \log \sum\_{j=0}^{K-1}\frac{ \pi\_\theta (y\_{<t}^{(j)}|x) }{\pi_{\theta\_\text{old}}(y\_{<t}^{(j)}|x)}$
>
> Where $y\_{<t}^{(w)}$ denotes the response achieving the maximum reward over the group.
>
> **(Q3)** $\alpha$ is the coefficient for KL divergence with $\pi\_{\theta\_\text{old} }$.
>  $\pi\_{\theta\_\text{old}}$ is the data-collecting policy, which is set at the beginning of every iteration as the learned policy in the previous iteration ($\pi\_{\theta\_\text{old}} \leftarrow {\pi\_\theta}$) (Algorithm 1, L 294.)
>
> This mechanism is comparable to PPO, where log-ratio clipping is derived from KL divergence between $\pi_\theta$ and $\pi_{\theta_{\text{old}}}$. Both approaches aim to prevent the new policy from drifting too far from the old one.
>
> Meanwhile, $\beta$ is the coefficient for KL with $\pi\_\text{init}$ (the initial basemodel). This term is commonly used in LLM post-training (including PPO).

---

> ### Comment · Reviewer_7Q1n · 2025-11-26
>
> Thank you again for the clarifications.
>
> >  $\alpha$ is the coefficient for KL divergence with $\pi_{\theta_\text{old} }$.  $\pi_{\theta_\text{old}}$ is the data-collecting policy, which is set at the beginning of every iteration as the learned policy in the previous iteration ($\pi_{\theta_\text{old}} \leftarrow {\pi_\theta}$) (Algorithm 1, L 294.)
>
> Then, I'm sorry, I completely misunderstood because when I read equation (1) I was thinking of the regularization towards $\pi_\text{init}$ as in [1], which, as you say, what is commonly done. But then, I think your formulation (and prior explanation) of equation (1) is misleading because you call $\pi_{old}$ the reference policy. Also, is it fair to say then that your true objective is something like:
>
> $E[R(x, y) - \alpha \log \frac{\pi_\theta(y|x)}{\pi_{old}(y|x)} - \beta \log \frac{\pi_\theta(y|x)}{\pi_{init}(y|x)}]$.
>
> [1] Stiennon et al. "Learning to summarize from human feedback" https://arxiv.org/abs/2009.01325

---

> > ### Author Response · Authors · 2025-11-26
> >
> > Yes, $E[R(x, y) - \alpha \log \frac{\pi_\theta(y|x)}{\pi_{old}(y|x)} - \beta \log \frac{\pi_\theta(y|x)}{\pi_{init}(y|x)}]$ is our true objective.
> >
> > Sorry for the confusion caused by referring to $\pi_{\theta\_\text{old}}$ as the reference policy. We will clarify that.

---

> ### Comment · Reviewer_7Q1n · 2025-11-26
>
> I also would like to echo some of qVzy's concerns. It's not clear to me what does training on prefixes buy you here. Your loss function could well be defined just on full sequences. Your own experiments show that using less than the full sequence actually damages performance (Q3 in the first batch, sorry for not keeping the numeration consistent afterwards). Similarly, as -I think- argued by qVzy, you could using a single stochastic rollout train on subsequences using plain policy gradients (even though that should just damage performance as you are throwing away the information from the suffix). So the fact that you train on prefixes starts to look more like an ablation to me than a feature of your method.

---

> ### Author Response · Authors · 2025-11-29
>
> Our method leverages the group interaction. It harnesses the K samples to assign the different weights to tokens within a trajectory by comparing partial sequences at every step.
> In the table above,
> $T$ indicates that we average over all time steps $t$, meaning we account for every possible prefix within the group.
> The results show that this setting achieves the highest performance, confirming the effectiveness of our group interaction strategy.
> Conversely, If we consider only the full sequence, we cannot utilize these intermediate interactions. As reported in the paper (Table 2, Ours(one-step)), the performance in this case drops (1.44).

---

### Official Review · Reviewer_SjqK · 2025-10-31

**Soundness:** 4
**Presentation:** 4
**Contribution:** 4
**Rating:** 8
**Confidence:** 4

**Summary:**

The paper introduces Group‑Normalized Implicit Value Optimization (GN‑IVO), a critic‑free RL fine‑tuning algorithm for LLMs. Starting from a KL‑regularized objective, the authors prove an explicit link between the prefix policy and a soft value. Then train values via a group‑normalized distributional matching objective, and finally eliminate the explicit value network.

**Strengths:**

The paper proposes a new loss that induces token/step‑level credit assignment without a critic by normalizing over groups of sampled completions and working with prefix policy ratios rather than explicit value estimates or partition functions.

Theorems 3.1–3.2 give a principled derivation: (i) an explicit policy–prefix‑value relationship; (ii) a consistency result that the group‑normalized objective recovers $V$ up to a constant; and (iii) policy invariance to additive shift.

The experiments are extensive and span both reasoning and open‑ended generation with multiple backbones.

**Weaknesses:**

The related‑work and experiments omit PRIME, which also leverages the policy to construct dense objectives. Although PRIME lacks the theoretical guarantees you present, it is methodologically close in spirit (dense reward objective constructed based on the implicit representation of the policy) and would be an informative reference and empirical comparison.


The objective presented use a uniform distribution sampling mechanism to compute the token level objective. Typically, the loss functions in RL average the loss including all tokens. It is not clear why this common approach is not used. Does this decision affect training? Can the author provide experiments results around this?

**Questions:**

Is there some kind of gradient detatch in the new loss function?

---

> ### Author Response · Authors · 2025-11-21
>
> **(Weakness 1) Comparison with PRIME**
>
> PRIME trains an additional reward model to provide dense signals by using binary cross entropy based on correctness. This approach’s performance is dependent on this learned reward model. Due to computational memory constraints, we used a small base model for the reward model in additional experiments. The results are as follows.
>
> *Reward models: Llama-3.2-1B, Qwen2.5-0.5B.*
> | Method                   | AMC23 P@1 | P@3 | MM P@1 | P@3 | OB P@1 | P@3 | AIME24 P@1 | P@3 | AIME25 P@1 | P@3 |
> |:-------------------------|----------:|----:|-------:|----:|-------:|----:|-----------:|----:|-----------:|----:|
> | Llama-3.1-8B-Instruct|          |    |       |    |       |    |           |    |           |    |
> | PRIME                    | 20.2      | 36.7| 15.7   |29.4 |16.7    |25.9 | 0.0        | 3.3| 0.0        | 0.0 |
> | ours                     | 42.5      | 45.0| 26.1   |36.0 |17.3    |27.8 |10.0        |16.6| 3.3        | 3.3 |
> | Qwen2.5-Math-7B      |          |    |       |    |       |    |           |    |           |    |
> | PRIME                    | 50.0      | 72.5| 26.1   |37.8 |37.7    |49.1 |26.6        |36.6|16.6        |20.0 |
> | ours                     | 62.5      | 75.0| 31.6   |41.9 |39.8    |49.0 |30.0        |40.0|13.3        |23.3 |
>
> Although PRIME trains an additional network to provide dense rewards, it is less effective than our method. This is because the reward model cannot provide sufficiently high-quality feedback.
>
> **(Weakness 2) Sampling $t$**
>
> Following your suggestion, we conducted experiments by varying the number of sampled time steps across {1, 20, $T$} (where $T$ means the average loss over $t$) in the HH-RLHF.
> The training curves are presented in Figure 7. (Appendix page 19)
>
> | # sampled steps \(t\) | 1 | 20 | T (full) |
> |:--------------------------:|:-----:|:-----:|:------------:|
> | Ours (score ↑)          | 1.40  | 1.65  | 1.80         |
>
> We observed that performance improves as the number of sampled $t$ increases.
>
> We sample $t$ 5 times for math reasoning and 20 times for text generation tasks.
> Thank you for the suggestion. While we initially focused only on uniform sampling, we have expanded our scope based on your feedback.
>
> **(Question 1) Gradient detach**
>
> We don’t use the gradient detach.

---

### Official Review · Reviewer_qVzy · 2025-11-01

**Soundness:** 2
**Presentation:** 3
**Contribution:** 2
**Rating:** 2
**Confidence:** 5

**Summary:**

This paper proposes a distribution matching approach to train a policy where the actions are tokens/steps in a natural language response. To achieve this, the authors generate multiple responses given a query and compute the reward  of each of the responses. A softmax over the rewards is computed to define the target distribution. A network is trained to predict the value at each token/step for each of the generated outputs. A softmax is applied over the outputs of the value network and cross-entropy loss between the softmaxed values and softmaxed rewards is minimized.

The authors show that the value function at any time-step can be parameterised using the ratio of \pi_\theta and \pi_old of the pastial sequence till that step. This allows training of the policy whose actions are tokens or steps without actually learning a value function or a stepwise reward model.

The results show reasonable performance improvement on Llama as well as Qwen models for Math as well as text benchmarks

**Strengths:**

- The primary novelty of the paper lies in using the relationship between the soft value function and prefix probability ratios for directly training the policy without the intermediate value function.
- The writing is crisp and clear.
- The results show reasonable gains on Math as well as general text generation benchmarks.

**Weaknesses:**

- The paper motivates fine-grained credit assignment, but Equation (9) assigns the same scalar return to every prefix of a sampled completion.
   - For example, if output y(1) earns higher terminal reward than y(2), the loss in (9) gives every prefix y(1)_{<t} higher weight than the same-length prefix y(2)_{<t}, regardless of which step actually caused the outcome difference. This is a Monte Carlo, every-visit return broadcast uniformly along the trajectory, not step-level credit assignment.
   - Accordingly, the claim that the method “solves credit assignment without a critic” is overstated. This does not imply the derivation around Eq. (7) is incorrect. Using a single sampled continuation to form a one-sample estimate of the soft value V(x, y_{<t}) is standard and, in expectation, corresponds to a weighted cross-entropy / distributional matching objective between the true and learned values. The issue arises when that one-sample value proxy is used directly to train the policy in Eq. (9): without a time-varying baseline/critic, the update collapses to sequence-level weighting replicated at all timesteps, providing no within-trajectory credit differentiation.

- Many of the ideas in the paper have been explored in some form or the other in other papers:
   - Distribution matching of self-normalized probability ratios with softmaxed rewards has already been explored in [1] (check equation 15 and equation 21)
   - The relationship between the soft value function and the probability of prefix of a trajectory has been explored in Eqs. (23)–(25) and (30)–(32) of [2].
  - The novelty of this current paper is to use the above relationship for directly training the policy without the intermediate value function.
- There are other works such as VinePPO[3] and SPO[4] that do Monte-Carlo simulations to estimate value functions without training a critic. These works are worth mentioning.

[1] BRAIn: Bayesian Reward-conditioned Amortized Inference for natural language generation from feedback
[2] Revisiting Maximum Entropy Inverse Reinforcement Learning: New Perspectives and Algorithms
[3] VinePPO
[4] SPO: Segment Policy Optimization

**Questions:**

- From Eq. (7) to Eq. (9): When you use the one-sample value estimate directly to weight policy updates at every timestep, what prevents the objective from collapsing to sequence-level weighting replicated across timesteps? Is there any mechanism that yields differential credit across steps of the same trajectory?
- The authors introduce a softmax after equation (9) claiming that it adds stability. Why wasn't it introduced in equation (7) itself in their intuitive objective?

---

> ### Author Response · Authors · 2025-11-21
>
> **(Weakness 1 & Question 1) Credit assignement in Eq.(9)**
>
> As noted in the review, Eq.(9) assigns the same reward to its corresponding prefix and does not perform Monte Carlo rollouts at every step. However, our method leverages the group interaction. Eq.(9) harnesses the K samples to provide the different credit to tokens within the same trajectory by comparing partial sequences at every step.
>
> For example, consider two sequences $y^{(1)}$ and $y^{(2)}$ within a group that obtain different rewards, $(r^{(1)}, r^{(2)})$ but share the same prefixes $y^{(1)}\_{<t}=y^{(2)}_{<t}$.
> The update for the shared prefix $y^{(1)}\_{<t}$ reflects the average reward over $(r^{(1)}, r^{(2)})$, while $y^{(1)}\_{t:T}$ is updated based solely on its reward $r^{(1)}$ to align with the soft values.
>
> In language tasks, generated sequences frequently shared identical prefixes. (For example, ‘Step1. Restate the problem…’ in reasoning tasks and ‘The author is considering …’ in summary tasks). To ensure precise every step-level credit assignment, standard approaches require Monte Carlo rollouts at every time step; however, this method incurs high sampling costs. Instead, our objective contrasts the prefixes within the group at every step to assign different credits to tokens efficiently.
>
>
> **(Weakness 2) Distinctions from BRAIN [1] and MaxEnt IRL [2]**
>
> We thank the reviewer for pointing out these references. We have updated the manuscript to include this.
>
> *(Weakness 2-1) Comparison with BRAIN [1]*
>
> We agree that combining Eq.(15) and Eq.(21) in [1] yields a formulation similar to our bandit-setting objective. However, we emphasize that our derivation and its underlying motivation are distinct from [1]. BRAIN [1] is primarily a policy-gradient method, where the advantage is defined as (self-normalized importance sampling - self-normalized baselines). In their framework, Eq.(15) represents the divergence between these self-normalized terms and is introduced specifically to theoretically justify the unbiasedness of their gradient estimator, rather than serving as the direct optimization objective.
> In contrast, our formulation uses this objective directly for optimization. We aim to implicitly estimate values via the policy, employing group normalization to cancel the intractable partition function. Furthermore, we note that BRAIN derives the softmax reward by assuming a Bradley-Terry model and requiring the proposal distribution to be the same as the prior. Our objective does not rely on any assumptions.
>
> *(Weakness 2-2) Relationship between the soft value and the probability of prefix*
>
> [2] also analyzes the relationship between prefixes and the full sequences, representing it as the expected exponential reward over future paths in an MDP.
> However, a crucial distinction exists due to the different problem formulations.
> In MaxEnt IRL [2], the value is an intrinsic property defined within a single probabilistic model.
> In contrast, our soft value function $V$ is derived from the optimality condition of a KL-regularized RL objective, which inherently involves two probabilistic models: $\pi^*\_\theta$ and $\pi\_{ \theta_{\text{old}} }$. Consequently, our $V$ fundamentally characterizes the relationship between these two policies, a formulation that naturally leads to the policy ratio derived in our Theorem 3.1.
>
> **(Weakness 3) Comparison with VinePPO and SPO**
>
> VinePPO and SPO address the credit assignment problem without relying on a critic. However, they rely on Monte Carlo rollouts at every time step to estimate values, leading to a sample complexity of O(K * T) per query. Although these methods successfully improve the performance, their sampling overhead is significantly higher than ours.
> We have also incorporated this comparison into the revised manuscript.
>
>
> **(Question 2)**
>
> We do not introduce the softmax within Eq.(7) because the linearity of expectation does not apply to the denominator.
>
> To be specific:
>
> $\frac{ e^{ V( x, y^{(i)}\_{<t} ) }}{  \sum\_j e^{ V( x, y^{(j)}\_{<t} ) } }
> = \frac{ E\_{\pi\_{ \theta\_\text{old}} (y|y\_{<t}) } [e^{ R( x, y^{(i)} ) } ] }{  \sum\_j E_{ \pi\_{\theta\_\text{old}} (y|y\_{<t}) } [ e^{ R( x, y^{(j)} ) } ]  } \neq E\_{\pi\_{ \theta\_\text{old}} (y|y\_{<t}) }  [ \frac{ e^{ R( x, y^{(i)} ) }}{  \sum\_j e^{ R( x, y^{(j)} ) } } ].
> (E[X]/E[Y] \neq E[X/Y])
> $
>
> Since we cannot simply move the expectation inside the softmax function, we introduce the softmax after Eq.(9).
>
> [1] BRAIn: Bayesian Reward-conditioned Amortized Inference for natural language generation from feedback
>
> [2] Revisiting Maximum Entropy Inverse Reinforcement Learning: New Perspectives and Algorithms

---

> ### Comment · Reviewer_qVzy · 2025-11-26
>
> Thanks for the response.
> 1) The authors mention that their approach allows for prefixes that are common across multiple rollouts to receive the average of the rewards. This is true in most on-policy RL algorithms  - including Policy Gradient. In Policy Gradient, if two rollouts share the same prefix, the effective reward received by the prefix is the average reward of the two prefixes.
> However, it would be unfair to say that Policy Gradient solves credit assignment.
> 2) In BRAIn, the combination of equation (15) and equation (22) is indeed the objective used in training. The gradient of that objective is equation (13). Also, BRAIN doesnt require the proposal distribution $q’$ to be same as prior. The computation of alphas and betas change according to the chosen proposal distribution. As shown in equation (21), the proposal is updated to the latest policy after every few updates.
> 3) I agree that VinePPO and SPO are more expensive, it would be good to see comparisons with them. They can serve as an upper bound to the performance.
> 4) Exclusion of softmax for proving something and then suddenly including it in the end "for stability" makes the proof questionable.

---

> > ### Author Response · Authors · 2025-11-29
> >
> > **Comparison VinePPO and SPO**
> >
> > We additionally evaluated VinPPO and SPO, and the results are presented below.
> >
> > | Method | AMC23 P@1 | P@3 | MM P@1 | P@3 |OB P@1 | P@3 | AIME24 P@1 | P@3 | AIME25 P@1 | P@3 |
> > |:---|---:|---:|---:|---:|---:|---:|---:|---:|---:|---:|
> > | Llama-3.1-8B-Instruct |  |  |  |  |  |  |  |  |  |  |
> > | VinePPO | 45.0 | 52.5 | 28.3 | 36.3 | 15.7 | 25.9 | 6.7 | 13.3 | 3.3 | 3.3 |
> > | ours         | 42.5 | 45.0 | 26.1 | 36.0 | 17.3 | 27.8 | 10.0 | 16.6 | 3.3 | 3.3 |
> > | Qwen2.5-Math-7B |  |  |  |  |  |  |  |  |  |  |
> > | VinePPO | 67.5 | 72.5 | 34.3 | 45.9 | 35.2 | 49.3 | 26.7 | 46.7 | 16.7 | 25.0 |
> > | SPO        | 60.0 | 75.0 | 34.3 | 39.5 | 20.0 | 36.6 | 10.0 | 20.0 | 16.3 | 23.3 |
> > | ours         | 62.5 | 75.0 | 31.6 | 41.9 | 39.8 | 49.0 | 30.0 | 40.0 | 13.3 | 23.3 |
> >
> > While VinePPO and SPO serves as performance upper bounds due to their O(K*T) sampling cost, our algorithm demonstrates comparable performance while only utilizing K samples.
> >
> > Regarding the SPO [1] experiment on Llama, we attempted the training but encountered significant technical issues. SPO determines segment boundaries based on token-level low log-probability regions. We observed that Llama exhibits consistently lower confidence across generated tokens. This caused SPO to generate an excessive number of fragmented segments, resulting in a drastic increase in sampling cost that exceeds our available resources.
> >
> > [1] (SPO) Segment Policy Optimization, Neurips 2025
> >
> > **Normalized exponential reward for stability via bounded gradients**
> >
> > While exponential terms tend to explode as values increase, by employing a normalized exponential reward, we obtain a gradient that is bounded.
> > Here, the target distribution is $p\_i = \frac{ e^{ R^{(i)} }}{ \sum\_j e^{R^{ (j) } } }$,
> > while the model distribution is $q\_i = \frac{  e^{\log{\frac{ \pi_{\theta}(y\_{<t}^{(i)}\mid x) }{ \pi_{\theta_\text{old}}(y\_{<t}^{(i)}\mid x)   }}} }{  \sum_j  e^{\log{\frac{\pi_{\theta} (y\_{<t}^{(j)} \mid x)  }{\pi_{\theta_\text{old}} (y_{<t}^{(j)}\mid x) }} }} $. The logits $z_i$ for the distribution $q$ are defined as the log-policy ratio:
> > $z\_i = \log \frac{ \pi\_\theta (y\_{<t}^{(i)}|x) }{ \pi\_{\theta\_\text{old}} (y\_{<t}^{(i)}|x) }$.
> > The gradient of the cross-entropy with respect to these logits is $p_i - q_i$.
> > This term is bounded because both $p$ and $q$ lie on the probability simplex.
> > Consequently, the normalized reward ensures training stability by preventing gradient explosion.

---

### Official Review · Reviewer_mHf2 · 2025-11-02

**Soundness:** 3
**Presentation:** 3
**Contribution:** 4
**Rating:** 8
**Confidence:** 3

**Summary:**

Reinforcement Learning has become central to LLM tuning and therefore RL research in this area has experimented a lot of interesting developments. To include some, the application of PPO to LLM RLHF, DPO and recently GRPO application to RLVR. This last one avoids the need for training a value network of PPO, leveraging instead empirical estimates using rollouts. Importantly, GRPO directly backpropagates the signal provided by the reward at the end of generation (thinking and answer) and has therefore no explicit credit attribution i.e. does not attempt to determine which steps (token generation, thinking step generation) have more relevance for the final reward observed.

The method proposed here also provides a well justified method to avoid having to train a separate value function while still obtaining step-level credit attribution. It also relies on empirical estimates as in GRPO.

The main idea is based on a learning objective that attempts to learn the value function from rollouts.

1. It proposes for this a distribution matching approach that can be seen as a weighted cross entropy loss between two energy models (Eq 7). One is constructed from the true value function and another is our parametrized value function that we want to learn.

2. In a similar vein as DPO, the optimal policy is here leveraged to express the parametrized value function in terms of the current and old policy, both eliminating the need for a separate network and providing a connection with the LLM that we want to train. This requires the formulation of the optimal policy at step level

**Strengths:**

1. The method seems, to the best of my understanding, correctly derived from well known principles and yields a very interesting formulation that achieves credit attribution while avoiding a separate value network and incorporating empirical estimates as GRPO.

2. Experimental setup is convincing and the results are strong across multiple categories including math and text generation. They also provide sensible baselines that also achieve step-level credit attribution through other formulations.

**Weaknesses:**

Only minor: some more results would further strengthen the paper i.e. having results for step-level vs non step-level for math would be interesting. For math, authors defined a step is as a reasoning step so having these results would be relevant.

Also since the samples are here used to describe distributions it is interesting to see if this makes it more dependent on a high K than e.g. GRPO. This could be added to the results in Figure 2.

**Questions:**

1. What are the one-step results for the math datasets in Table 1? why not include these since they are included in the rest of the experiments (maybe I missed something). Also what is the effect of K on alternative methods like GRPO?.

2. Please also explain how the findings and stated theorems relate to the findings of [2]. In particular how Theorem 3.1 relates to the result from (Ziebart, 2010) used in Eq 5.

[2] From r to Q∗: Your Language Model is Secretly a Q-Function

---

> ### Author Response · Authors · 2025-11-21
>
> **(Weakness 1) Single step-level in math reasoning**
>
> We report the single-step performance for math reasoning in the below table.
> Generally, as observed in the task generation tasks, our bandit objective demonstrates limited performance compared to the step-level approach.
>
> | Method          | AMC23      |        | MM         |        | OB         |        | AIME24     |        | AIME25     |        |
> |:----------------|----------:|------:|----------:|------:|----------:|------:|----------:|-----:|---------:|----:|
> |    | P@32       | P@128  | P@32       | P@128  | P@32       | P@128  | P@32       | P@128  | P@32       | P@128  |
> | Llama-3.1-8B-Instruct    |           |     |        |     |        |     |            |     |            |     |
> | ours (one-step)        | 35.0      | 40.0| 25.3   |34.1 |17.9    |26.1 |10.0        |13.3 | 0.0        | 3.3 |
> | ours                   | 42.5      | 45.0| 26.1   |36.0 |17.3    |27.8 |10.0        |16.6 | 3.3        | 3.3 |
> | Qwen2.5-Math-7B    |           |     |        |     |        |     |            |     |            |     |
> | ours (one-step)        | 57.5      | 70.0| 31.2   |41.9 |38.5    |49.4 |23.3        |30.3 |13.3        |13.3 |
> | ours                   | 62.5      | 75.0| 31.6   |41.9 |39.8    |49.0 |30.0        |40.0 |13.3        |23.3 |
>
> **(Weakness 2 & Question 1) Comparison of K dependency with GRPO**
>
> To investigate the dependency on group size $K$ relative to GRPO, we conducted additional experiments varying K for GRPO. The results, shown in Figure 2, indicate that $K$ is also the key hyperparameter in GRPO; its performance improves as K increases.
> Notably, when the group size is small (especially $K=2$), the performance gap between our method and GRPO is larger than at  larger group sizes. This demonstrates that our algorithm is effective even in the small group size.
>
> | Model/Method | K=2 | K=4 | K=8 | K=16|
> |:---|---:|---:|---:|---:|
> | Qwen2.5-1.5B-Instruct | | | | |
> | GRPO | 1.2 | 1.5 | 1.6 | 1.8 |
> | ours | 1.4 | 1.6 | 1.7 | 2.0 |
>
>
> **(Question 2) Relation to findings in [1]**
>
> Based on our findings that $\pi\_{\theta^\star} (y\_{<t}|x) = \pi\_{\theta_\text{old}}(y\_{<t}|x)e^{V(x,y\_{<t})} / Z(x)$ , we derive the following:
>
> $ \frac{\pi\_{\theta^\star} (y\_{<t+1}| x) }{ \pi\_{\theta^\star} (y\_{<t}|x) }( =\frac{\pi\_{\theta^\star}(y\_{t}, y\_{<t} | x) }{\pi\_{\theta^\star}( y\_{<t}|x ) })
> = \frac{\pi\_{ \theta_\text{old} } (y\_{<t+1}|x) } { \pi\_{\theta\_\text{old}} (y\_{<t}|x) } \frac{ e^{ V(x,y\_{<t+1})} }{ e^{V(x,y\_{<t})} }$.
>
> $\rightarrow \pi\_{\theta^\star} (y\_{t}|x, y\_{<t}) = \pi\_{ \theta\_\text{old} } (y\_{t}|x, y\_{<t})
> \frac{ e^{V(x,y\_{<t+1}) } }{ e^{ V( x,y\_{<t} ) } }
> =  \pi\_{ \theta\_\text{old} } (y\_{t}|x, y\_{<t}) \exp{ (\underbrace{V(x,y\_{<t+1})}_{ Q(x, y\_{<t}, y_t) } - V( x,y\_{<t} ) ) }$.
>
> This result aligns with Eq.(5) in [1].
>
>
> [1] estabilishes the relationship between $\pi\_{\theta^\star} (y\_{t}|x, y\_{<t})$ and the reward by summation over time steps $t$.
> Similary, we sum the log-probability ratios over t (where $\log\frac{ \pi\_{\theta^\star} (y\_t |x, y\_{<t})}{ \pi\_{ \theta\_\text{old} } (y\_t | x, y\_{<t})} = V(x, y\_{<t+1} - V(x, y\_{<t})$), we observe a telescoping sum:
>
> $
> \sum_t \log\frac{ \pi\_{\theta^\star} (y\_t |x, y\_{<t})}{ \pi\_{ \theta\_\text{old} } (y\_t | x, y\_{<t})} = \sum_t V(x, y\_{<t+1}) - V(x, y\_{<t}) = R(x, y)/\alpha - V(x) $.
>
> This derivation confirms that the finding in [1] is consistent with our framework.
>
> [1] From r to Q^*: Your Language model is secretly a Q-function

---

### Meta-Review · Area_Chair_bHCC · 2026-01-01

**Summary:**

This paper introduces a critic-free RL fine-tuning approach for LLMs that learns implicit step values via a group-normalized distribution matching objective and connects these values to prefix policy ratios. Reviewers found the approach well-motivated and theoretically grounded, with strong empirical results across both reasoning and open-ended generation tasks. One reviewer raised serious concerns about whether the method truly provides within-trajectory credit assignment, which has to be addressed in the final version of the paper by additional supporting evidence. The overall reception is positive and the authors provided substantial additional experiments and clarifications in the rebuttal. I recommend acceptance.

**Reviewer Concerns:**

Addressed in rebuttal:
- Added one-step bandit math results.
- Added key baselines.
- Added evaluation with an LLM-judge.
- Clarified implementation details (step sampling counts, Pass@128).
- Acknowledged the need for wording clarity around the “reference policy” terminology.

Still outstanding:
- A core disagreement remains on whether the method provides true within-trajectory credit assignment vs. sequence-level weighting propagated to prefixes; this calls for claim calibration and clearer framing of what “step-level” means here.
- Related-work positioning could be improved in the camera-ready version of the paper.

**Reviewer Scores:**

mHf2 (8): likely remain unchanged.

SjqK (8): likely remain unchanged.

7Q1n (6): Likely remain unchanged.

qVzy (2): likely remain unchanged.

---

### Decision · Program_Chairs · 2026-01-26

Accept (Poster)